# Physics-Informed Generative Modeling of Wireless Channels

**Benedikt Böck** [1]   **Andreas Oeldemann** [2]   **Timo Mayer** [2]   **Francesco Rossetto** [2]   **Wolfgang Utschick** [1]

## Abstract

Learning the site-specific distribution of the wireless channel within a particular environment of interest is essential to exploit the full potential of machine learning (ML) for wireless communications and radar applications. Generative modeling offers a promising framework to address this problem. However, existing approaches pose unresolved challenges, including the need for high-quality training data, limited generalizability, and a lack of physical interpretability. To address these issues, we combine the physics-related compressibility of wireless channels with generative modeling, in particular, sparse Bayesian generative modeling (SBGM), to learn the distribution of the underlying physical channel parameters. By leveraging the sparsity-inducing characteristics of SBGM, our methods can learn from compressed observations received by an access point (AP) during default online operation. Moreover, they are physically interpretable and generalize over system configurations without requiring retraining.

## 1. Introduction

The accurate modeling of wireless channels is critical for system design, network optimization, and performance evaluation in wireless communications and radar (Wang et al., 2018; 2020; Yin & Cheng, 2016). There are two approaches to wireless channel modeling: physically and analytically oriented (Almers et al., 2007). The former integrates the underlying physics in the description of wireless channels and focuses on their accuracy and realism. In contrast, the latter captures channel statistics, ignoring the underlying physics, and provides an easy-to-use framework for evaluation and system design schemes (Almers et al., 2007; Imoize et al., 2021). Standardized channel models such as the 3GPP (3GPP, 2024a), COST (Liu et al., 2012) or WINNER

(Meinilä et al., 2009) families are physically oriented and combine physics with statistical channel properties. These models use the laws of electromagnetic wave propagation to describe channels as a function of the corresponding physical parameters (directions of arrival (DoAs), delays, etc.), which in turn are characterized statistically. These characteristics are calibrated and evaluated by real-world measurement campaigns in different propagation scenarios such as indoor, outdoor, urban, and rural (Yin & Cheng, 2016). While these models characterize channels in generic scenarios and are widely accepted in the communication community, they also exhibit considerable drawbacks.

Due to their broad categorization into generic scenario types, such as indoor and outdoor, these channel models cannot represent the site-specific characteristics of particular environments. This limits their use in ML-aided wireless communications, where site-specific channel realizations are critical for training (Kim et al., 2023). The emerging variety of use cases in communications, such as millimeter wave, unmanned aerial vehicles, high-speed train, ultra-massive multiple-input-multiple-output (MIMO), joint communications and sensing, and Internet of Things communications, all exhibit their own unique requirements for accurate channel modeling and impose open challenges for measurement campaigns and channel sounders (Wang et al., 2020). Additionally, the improved accuracy of physically oriented channel models over time made it more difficult to directly use them in real-time processing tasks, resulting in an undesired accuracy-simplicity trade-off (Imoize et al., 2021).

One physically oriented alternative is ray tracing, which models the channel purely deterministically. Ray tracing is a technique to simulate the electromagnetic wave propagation in specific scenarios (Hofmann, 1990). Ray tracing for channel modeling requires a 3D digital replica of the scenario and an accurate characterization of the materials within the scene (Geok et al., 2018; McKown & Hamilton, 1991; Yun & Iskander, 2015). Commercial ray tracing tools like WirelessInside (Remcom) use databases to determine the material properties, requiring full environmental knowledge. To address this limitation, differentiable ray tracing is studied in (Gan et al., 2014; Hoydis et al., 2024; Orekondy et al., 2023), enabling the learning of material properties from data. On the downside, high-quality channel state information (CSI) must be acquired for training in each

[1]Technical University of Munich, Munich, Germany [2]Rohde & Schwarz, Munich, Germany. Correspondence to: Benedikt Böck <benedikt.boeck@tum.de>.

*Proceedings of the 42nd International Conference on Machine Learning*, Vancouver, Canada. PMLR 267, 2025. Copyright 2025 by the author(s).

scenario of interest, questioning its practicality. Additionally, modeling diffuse scattering and the complexity remain challenging objectives in ray tracing (Imoize et al., 2021).

Another approach for channel modeling is based on generative modeling, which aims to learn an unknown distribution from data (Bond-Taylor et al., 2022). Using the terminology from (Diggle & Gratton, 1984), generative models can be categorized into *prescribed* and *implicit* models. *Prescribed* models learn the parameters of a statistical model (Girin et al., 2021). In contrast, *implicit* models directly learn to generate samples (Mohamed & Lakshminarayanan, 2017). Examples of the former are variational autoencoders (VAEs) (Kingma & Welling, 2014), while generative adversarial networks (GANs) (Goodfellow et al., 2014) represent the latter. Perhaps surprisingly, almost all current approaches to modeling wireless channels with generative models use GANs and, thus, rely on *implicit* modeling (Euchner et al., 2024; Hu et al., 2023; Li et al., 2018; Orekondy et al., 2022; Seyedsalehi et al., 2019; Tian et al., 2024; Xiao et al., 2022; Yang et al., 2019). Moreover, since this line of research applies GANs in a black box manner, these methods belong to the analytically oriented branch of channel models. While these publications show some progress in channel modeling with *implicit* generative models, there are several open concerns with this approach (Euchner et al., 2024):

*High-quality datasets*: The assumption of having access to (lots of) site-specific high-quality training data requires costly measurement campaigns in each scenario of interest.

*Generalizability*: The proposed GAN-based methods only generate channel realizations matching the system configuration (number of antennas, subcarrier spacing etc.) used for training, making it difficult to adapt to other configurations.

*Physical interpretability & consistency:* Their generation process cannot be interpreted physically and they cannot guarantee their samples to obey the underlying physics.

In this work, we demonstrate how *prescribed* generative models can effectively resolve all these limitations. Specifically, we show that the parameterized statistical models in *prescribed* generative modeling enable integrating the physics of electromagnetic wave propagation. This adaptation not only relaxes the requirements for the training data but also shifts the generative modeling approach from the analytically oriented to the physically oriented channel models and provides generalizability, interpretability, and physical consistency. Our approach shares features with standardized channel models, as it also describes the wireless channel as a function of the physical parameters. However, the statistical model that characterizes these parameters does not need to be calibrated through extensive and costly measurement campaigns but can be trained by a few compressed and noisy channel observations that an AP or a base station (BS)

receives during default online operation. Our method aligns with the idea of physics-informed ML, i.e., facilitating the learning from training data by incorporating underlying pre-known physics that the trained model must obey (Karniadakis et al., 2021; Raissi et al., 2019). However, instead of integrating partial differential equations from physics into a loss function, we leverage the laws of ray optics, building upon electromagnetic wave propagation and relating the channel to its physical parameters (e.g., delays).

**Main Contributions** We combine the physics-related compressibility of wireless channels with both *implicit* and *prescribed* generative models. The resulting models overcome limitations of existing approaches, i.e., they do not need high-quality training data, generalize over system configurations without requiring retraining, and are physically interpretable. By leveraging pre-known structural knowledge about conditional channel moments, we additionally show that the *prescribed* approach, in particular SBGM, provides advantages over the *implicit* approach by promoting sparsity and ensuring physical consistency. Moreover, we validate the performance on several datasets, showing the superiority of SBGM for parameter and channel generation.[1]

## 2. Related Work

Early work exploiting *implicit* GAN-based generative models for channel modeling is given by (Li et al., 2018; Yang et al., 2019). Since then, several GAN variants have been proposed for time-varying (Seyedsalehi et al., 2019) or MIMO channel impulse responses (Orekondy et al., 2022; Xiao et al., 2022). For evaluation, (Orekondy et al., 2022; Seyedsalehi et al., 2019; Xiao et al., 2022) use cluster delay line, tap delay line, or pedestrian A-like 3GPP channel models (3GPP, 2023; 2024a). These link-level models produce channels that all originate from the same cluster angles and delays and cannot represent proper communication scenarios where users experience different sets of delays, DoAs, and directions of depature (DoDs). The work in (Euchner et al., 2024; Hu et al., 2023) also studies GANs for MIMO channel impulse responses, but evaluates on the geometry-based stochastic channel model QuaDRiGa (Jaeckel et al., 2014) or measurement data. Diffusion models (DMs) for MIMO channels are studied in (Lee et al., 2024; Sengupta et al., 2023). The work (Tian et al., 2024; Xia et al., 2022) considers parameter generation (e.g., delays) using a ground-truth training dataset of parameters. In (Baur et al., 2024b; Fesl et al., 2023), Gaussian mixture models (GMMs) and VAEs are used for channel estimation based on compressed training data. Moreover, (Baur et al., 2025) introduces techniques to evaluate generative models for wireless channels.

---

[1]Source code is available at https://github.com/beneboeck/phy-inf-gen-mod-wireless.

## 3. Background and Problem Statement

### 3.1. Physically Characterizing Wireless Channels

Wireless signal transmission causes the signal to attenuate and undergo phase shifts. Moreover, as the signal propagates, it can also get reflected or scattered by obstacles. This results in multiple paths (or rays) between the transmitter and receiver, each characterized by its own attenuation and phase shift, which, in turn, are linked to the channel parameters (e.g., angles and delays) through geometrical optics. The wireless signal transmission can be modeled as a linear time-variant system (Tse & Viswanath, 2005) and, thus, these effects are captured by the wireless baseband channel (transfer function), which depends on the time $t$, the (baseband) frequency $f$, and the receiver and transmitter positions $\boldsymbol{r}_\mathrm{R}$ and $\boldsymbol{r}_\mathrm{T}$ in local coordinate systems, i.e.,

$$h(t, f, \boldsymbol{r}_\mathrm{R}, \boldsymbol{r}_\mathrm{T}) = \sum_{\ell=1}^{L} \sum_{m=1}^{M_\ell} \rho_{\ell,m} \, \mathrm{e}^{\mathrm{j}\, 2\pi \vartheta_{\ell,m} t} \, \mathrm{e}^{-\mathrm{j}\, 2\pi \tau_{\ell,m} f} \times$$
$$\mathrm{e}^{-\mathrm{j}\, \boldsymbol{k}(\boldsymbol{\Omega}_{\ell,m}^{(\mathrm{R})})^\mathrm{T} \boldsymbol{r}_\mathrm{R}} \, \mathrm{e}^{-\mathrm{j}\, \boldsymbol{k}(\boldsymbol{\Omega}_{\ell,m}^{(\mathrm{T})})^\mathrm{T} \boldsymbol{r}_\mathrm{T}} \,. \tag{1}$$

The channel parameters contain the number $L$ of paths, the number $M_\ell$ of subpaths per path $\ell$, the complex path losses $\rho_{\ell,m}$, the doppler shifts $\vartheta_{\ell,m}$, the delays $\tau_{\ell,m}$, the DoAs $\boldsymbol{\Omega}_{\ell,m}^{(\mathrm{R})}$, and the DoDs $\boldsymbol{\Omega}_{\ell,m}^{(\mathrm{T})}$. A schematic of the paths when transmitting wireless signals is given in Fig. 1. The wavevector $\boldsymbol{k}(\cdot)$ is defined as $\boldsymbol{k}(\cdot) = (2\pi/\lambda)\, \mathbf{e}(\cdot)$ with wavelength $\lambda$ and $\mathbf{e}(\cdot)$ is the spherical unit vector. Transforming (1) via a Fourier transform (FT) with respect to any of its arguments results in their dual so-called dispersive domains (Yin & Cheng, 2016). Moreover, one is typically interested in a sampled version of the wireless channel $h(\cdot)$. For instance, when the receiver is equipped with multiple antennas, receiving signals can be viewed as sampling the wireless channel $h(\cdot)$ in (1) at the antennas' spatial positions. By exemplary ignoring the channel's time- and frequency-dependency (i.e., $t = f = 0$) and accounting for the spatial sampling via multiple antennas at the receiver, (1) can be represented as

$$\boldsymbol{h} = \sum_{\ell=1}^{L} \sum_{m=1}^{M_\ell} \rho_{\ell,m} \boldsymbol{a}_\mathrm{R}(\boldsymbol{\Omega}_{\ell,m}^{(\mathrm{R})}) \tag{2}$$

with $\boldsymbol{a}_\mathrm{R}(\cdot)\big|_i = \mathrm{e}^{-\mathrm{j}\, \boldsymbol{k}(\cdot)^\mathrm{T} \boldsymbol{r}_{\mathrm{R},i}}$ and the position $\boldsymbol{r}_{\mathrm{R},i}$ of the $i$th receiving antenna. The vectors $\boldsymbol{a}_\mathrm{R}(\cdot)$ are typically referred to as steering vectors. Equation (2) represents the generic single-input-multiple-output (SIMO) channel, which does not cover the time and frequency domains. As another example, the receiver and transmitter in orthogonal-frequency-division-multiplexing (OFDM) are both equipped with single antennas, whose positions can be set to the coordinate systems' origins, i.e., $\boldsymbol{r}_{\mathrm{T/R},1} = \boldsymbol{0}$. Thus, OFDM covers no spatial domains and is characterized by its subcarrier spacing $\Delta f$ and symbol duration $\Delta T$. This setup corresponds

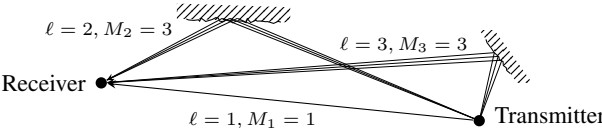

*Figure 1.* Path schematic when transmitting wireless signals.

to equidistantly sampling (1) in both the frequency and time domains, yielding the OFDM channel matrix

$$\boldsymbol{H} = \sum_{\ell=1}^{L} \sum_{m=1}^{M_\ell} \rho_{\ell,m} \boldsymbol{a}_t(\vartheta_{\ell,m}) \boldsymbol{a}_f(\tau_{\ell,m})^\mathrm{T} \tag{3}$$

with $\boldsymbol{a}_t(\vartheta_{\ell,m})\big|_i = \mathrm{e}^{\mathrm{j}\, 2\pi \vartheta_{\ell,m}(i-1)\Delta T}$ and $\boldsymbol{a}_f(\tau_{\ell,m})\big|_j = \mathrm{e}^{-\mathrm{j}\, 2\pi \tau_{\ell,m}(j-1)\Delta f}$. For our following considerations, we define the set of physical channel parameters

$$\mathcal{P} = \{\rho_{\ell,m}, \vartheta_{\ell,m}, \tau_{\ell,m}, \boldsymbol{\Omega}_{\ell,m}^{(\mathrm{R})}, \boldsymbol{\Omega}_{\ell,m}^{(\mathrm{T})}\}_{\ell,m=1}^{L,M_\ell}. \tag{4}$$

We specify subsets containing the parameters associated with a specific domain by their subscript, e.g., $\mathcal{P}_\mathrm{R} = \{\rho_{\ell,m}, \boldsymbol{\Omega}_{\ell,m}^{(\mathrm{R})}\}_{\ell,m=1}^{L,M_\ell}$ or $\mathcal{P}_{t,f} = \{\rho_{\ell,m}, \vartheta_{\ell,m}, \tau_{\ell,m}\}_{\ell,m=1}^{L,M_\ell}$.

### 3.2. Sparse Representation of Wireless Channels

One possibility to beneficially represent channels is by exploiting their compressibility (i.e., approximate sparseness) regarding a physically interpretable dictionary (Dai & So, 2021; Gaudio et al., 2022). For instance, by placing the receiving antennas with $\lambda/2$ spacing in a uniform linear array (ULA), aligning the local coordinate system's z-axis with the ULA, and assuming the ULA to be in the far-field, the DoAs $\boldsymbol{\Omega}_{\ell,m}^{(\mathrm{R})}$ reduce to position-independent angles $\omega_{\ell,m}^{(\mathrm{R})} \in [-\pi/2, \pi/2]$ with $\boldsymbol{k}(\cdot)^\mathrm{T} \boldsymbol{r}_{\mathrm{R},i} = \pi(i-1)\sin(\cdot)$ in (2). By additionally defining a grid $\mathcal{G}_\mathrm{R} = \{g\pi/S_\mathrm{R}\}_{g=-S_\mathrm{R}/2}^{S_\mathrm{R}/2-1}$ of cardinality $S_\mathrm{R}$, (2) can be represented by

$$\boldsymbol{h} = \sum_{g=-S_\mathrm{R}/2}^{S_\mathrm{R}/2-1} s_\mathrm{R}^{(g)} \boldsymbol{a}_\mathrm{R}\left(\frac{g\pi}{S_\mathrm{R}}\right) = \boldsymbol{D}_\mathrm{R} \boldsymbol{s}_\mathrm{R} \tag{5}$$

with $\boldsymbol{D}_\mathrm{R} = [\boldsymbol{a}_\mathrm{R}(-\frac{\pi}{2}), \ldots, \boldsymbol{a}_\mathrm{R}(\frac{\pi}{2} - \frac{\pi}{S_\mathrm{R}})]$, and $\boldsymbol{s}_\mathrm{R} = [s_\mathrm{R}^{(-S_\mathrm{R}/2)}, \ldots, s_\mathrm{R}^{(S_\mathrm{R}/2-1)}]^\mathrm{T}$. If the DoAs $\omega_{\ell,m}^{(\mathrm{R})}$ coincide with gridpoints in $\mathcal{G}_\mathrm{R}$, the entries $s_\mathrm{R}^{(g)}$ fulfill $s_\mathrm{R}^{(g)} = \rho_{\ell,m}$ if $\omega_{\ell,m}^{(\mathrm{R})} = g\pi/S_\mathrm{R}$ and are zero else. In this case, $\boldsymbol{s}_\mathrm{R}$ perfectly determines the channel's parameters $\mathcal{P}_\mathrm{R}$. Since this usually only approximately holds, $\boldsymbol{s}_\mathrm{R}$ is not exactly sparse but rather compressible. Equivalent compressible representations exist in the frequency and temporal domains. By defining the grid $\mathcal{G}_t \times \mathcal{G}_f$ with $\mathcal{G}_t = \{i2\overline{\vartheta}/S_t\}_{i=-S_t/2}^{S_t/2-1}$ and $\mathcal{G}_f = \{j\overline{\tau}/S_f\}_{j=0}^{S_f-1}$, where $\overline{\vartheta}$ and $\overline{\tau}$ bound the maximally reachable doppler shift and delay, and $S_t, S_f \in \mathbb{N}$, we also find a compressible representation of the vectorized OFDM channel in (3) equivalent to (5), i.e., $\boldsymbol{h} = \boldsymbol{D}_{t,f} \boldsymbol{s}_{t,f}$ with $\boldsymbol{D}_{t,f} = \boldsymbol{D}_t \otimes \boldsymbol{D}_f$. Appendix A.1 provides a more detailed description.

## 3.3. Generative Learning through Inverse Problems

In some cases, generative models should learn a data distribution that is only indirectly observable from the training data, e.g., if only compressed training data is available.

**Implicit Approach**  The GAN variant AmbientGAN is an *implicit* generative model that addresses this objective (Bora et al., 2018). GANs simultaneously train two neural networks (NNs), the generator $\mathcal{G}_{\boldsymbol{\theta}}(\cdot)$ and discriminator $\mathcal{D}_{\boldsymbol{\phi}}(\cdot)$, in a competitive setting (Goodfellow et al., 2014). The generator takes inputs $\boldsymbol{z}_i$ drawn from a fixed distribution $p(\boldsymbol{z})$ and aims to minimize a certain objective by producing samples $\mathcal{G}_{\boldsymbol{\theta}}(\boldsymbol{z}_i)$ that resemble the training samples $\boldsymbol{y}_i$. In ordinary GANs, the discriminator's goal is to differentiate directly between the generator's outputs and the training samples by maximizing the same objective. However, AmbientGAN assumes a (stochastic) mapping $\boldsymbol{f}(\cdot)$ that relates the training samples and the random variable $\boldsymbol{s}$ of interest (i.e., $\boldsymbol{y}_i = \boldsymbol{f}(\boldsymbol{s}_i)$) and applies $\boldsymbol{f}(\cdot)$ (i.e., $\boldsymbol{f}(\mathcal{G}_{\boldsymbol{\theta}}(\boldsymbol{z}_i))$) before forwarding the result to $\mathcal{D}_{\boldsymbol{\phi}}(\cdot)$. This trains the generator to generate samples $\boldsymbol{s}_i$ while only having $\boldsymbol{y}_i$ for training.

**Prescribed Approach**  SBGM is a *prescribed* generative modeling framework that also addresses the learning of hidden distributions (Böck et al., 2024b). It can learn a nontrivial parameterized distribution $p_{\boldsymbol{\theta},\boldsymbol{\delta}}(\boldsymbol{s})$ for the compressible representation $\boldsymbol{s}$ of a signal $\boldsymbol{h}$ of interest given a dictionary $\boldsymbol{D}$, i.e., $\boldsymbol{h} = \boldsymbol{D}\boldsymbol{s}$. This learning can be done solely by noisy and compressed training samples $\boldsymbol{y}_i = \boldsymbol{A}\boldsymbol{h}_i + \boldsymbol{n}_i$ with additive white Gaussian noise (AWGN) $\boldsymbol{n}_i$ and known measurement matrix $\boldsymbol{A}$. SBGM combines sparse Bayesian learning (SBL) (Wipf & Rao, 2004) with a sub-class of *prescribed* models, namely, conditionally Gaussian latent models (CGLMs), i.e., models with a conditioned Gaussian on a latent variable $\boldsymbol{z}$. After minorly extending it to complex numbers, the statistical model in SBGM is given by

$$\boldsymbol{y}|\boldsymbol{s} \sim p(\boldsymbol{y}|\boldsymbol{s}) = \mathcal{N}_{\mathbb{C}}(\boldsymbol{y}; \boldsymbol{A}\boldsymbol{D}\boldsymbol{s}, \sigma^2\,\mathbf{I}), \quad (6)$$

$$\boldsymbol{s}|\boldsymbol{z} \sim p_{\boldsymbol{\theta}}(\boldsymbol{s}|\boldsymbol{z}) = \mathcal{N}_{\mathbb{C}}(\boldsymbol{s}; \mathbf{0}, \mathrm{diag}(\boldsymbol{\gamma}_{\boldsymbol{\theta}}(\boldsymbol{z}))), \quad (7)$$

$$\boldsymbol{z} \sim p_{\boldsymbol{\delta}}(\boldsymbol{z}) \quad (8)$$

with known measurement matrix $\boldsymbol{A}$, learnable parameters $\boldsymbol{\theta}$ and $\boldsymbol{\delta}$, variance $\sigma^2$ of the measurement noise, latent variable $\boldsymbol{z}$, and $\boldsymbol{s}$ follows a conditionally zero-mean Gaussian $p_{\boldsymbol{\theta}}(\boldsymbol{s}|\boldsymbol{z})$ with $\boldsymbol{z}$-dependent diagonal covariance matrix. The work in (Böck et al., 2024b) shows that this particular choice of $p_{\boldsymbol{\theta}}(\boldsymbol{s}|\boldsymbol{z})$ serves as a sparsity-promoting regularization. It introduces the compressive sensing GMM (CSGMM) and compressive sensing VAE (CSVAE) as two particular implementations of SBGM. In the latter, a NN decoder outputs the conditional variances $\boldsymbol{\gamma}_{\boldsymbol{\theta}}(\cdot)$ revealing its *prescribed* characteristic. After training, $p_{\boldsymbol{\delta},\boldsymbol{\theta}}(\boldsymbol{s})$ is used for regularizing inverse problems and has been demonstrated to outperform standard priors in compressive sensing (Böck et al., 2024b).

## 3.4. System Models and Problem Statement

We consider a scenario in which an AP receives channel observations $\boldsymbol{y}_i$ from different locations within the environment it serves. One example is outdoor communications, where users periodically send pilot symbols to their BS. The observations are noisy and potentially compressed, i.e.,

$$\boldsymbol{y} = \boldsymbol{A}\boldsymbol{h} + \boldsymbol{n} \quad (9)$$

where $\boldsymbol{h}$ represents the channel, $\boldsymbol{A}$ is the measurement matrix, and $\boldsymbol{n}$ is AWGN. The AP eventually collects a dataset $\mathcal{Y} = \{\boldsymbol{y}_i\}_{i=1}^{N_t}$ that captures site-specific information from the entire environment. In our work, we focus on the following two systems.

*Narrowband static SIMO:* The receiver is equipped with multiple antennas, and the transmitter has a single antenna. The channel $\boldsymbol{h}$ only covers the spatial receiver domain and equals (2). The matrix $\boldsymbol{A}$ in (9) is an identity, i.e., $\boldsymbol{A} = \mathbf{I}_{M_{\mathrm{R}}}$.

*Double-selective OFDM:* The channel $\boldsymbol{h}$ in (9) covers no spatial domain, and vectorizes (3), i.e., $\boldsymbol{h} = \mathrm{vec}(\boldsymbol{H})$. The observation matrix $\boldsymbol{A}$ is a selection matrix, i.e., its rows are distinct unit vectors, masking out particular entries of $\boldsymbol{h}$.

As the channel $\boldsymbol{h}$ does not cover all four possible domains in these systems, only a subset of $\mathcal{P}$ in (4) determines $\boldsymbol{h}$. Thus, depending on the employed system, the environment is characterized by a distribution over this subset of physical parameters, i.e., $p(\mathcal{P}_{\mathrm{R}})$ in SIMO and $p(\mathcal{P}_{t,f})$ in OFDM. This work aims to develop a physics-informed generative model that can learn these distributions solely using the imperfect training dataset $\mathcal{Y}$. While this work focuses on SIMO and OFDM, there is a straightforward generalization to other systems, such as MIMO and MIMO-OFDM.

# 4. Physics-Informed Generative Approaches

Building on the sparse channel representation explained in Section 3.2 we observe that choosing a parameter grid $\mathcal{G}_{\mathrm{R}}$ or $\mathcal{G}_t \times \mathcal{G}_f$ with sufficiently high resolution enables the compressible channel representation $\boldsymbol{s}_{\mathrm{R}}$ or $\boldsymbol{s}_{t,f}$ to uniquely determine the channel's parameters $\mathcal{P}_{\mathrm{R}}$ or $\mathcal{P}_{t,f}$. Thus, we can replace the learning of $p(\mathcal{P}_{\mathrm{R}})$ and $p(\mathcal{P}_{t,f})$ by instead learning $p(\boldsymbol{s}_{\mathrm{R}})$ and $p(\boldsymbol{s}_{t,f})$, with the grid resolution being the hyperparameter that controls the approximation error.[2] Moreover, by incorporating the sparse channel representation into the dataset in Section 3.4, we reformulate our problem to that of learning $p(\boldsymbol{s})$ given $\mathcal{Y} = \{\boldsymbol{y}_i = \boldsymbol{A}\boldsymbol{D}\boldsymbol{s}_i + \boldsymbol{n}_i\}_{i=1}^{N_t}$ with $\boldsymbol{s}_i \sim p(\boldsymbol{s})$, and $\boldsymbol{n}_i \sim \mathcal{N}_{\mathbb{C}}(\mathbf{0}, \sigma_i^2\,\mathbf{I})$.[3]

---

[2]For readability, we omit the subscripts of $\boldsymbol{s}$, $\mathcal{P}$, and $\mathcal{G}$ from now on when the argument applies to both OFDM and SIMO.

[3]While the actual noise variance mainly depends on the receiver hardware and, thus, does not vary between users at different locations, the typical channel-wise pre-processing of normalizing by an estimated path loss effectively results in varying noise variances.

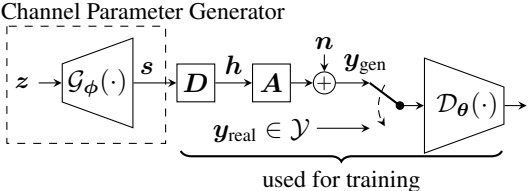

*Figure 2.* Layout of the physics-informed *implicit* AmbientGAN.

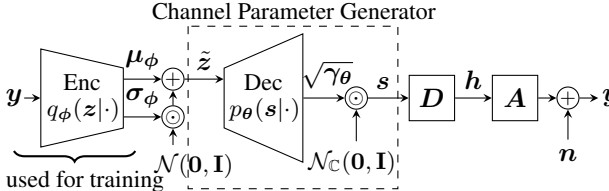

*Figure 3.* Layout of the physics-informed *prescribed* CSVAE.

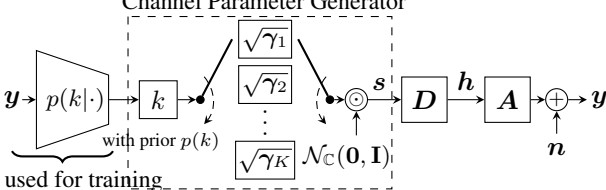

*Figure 4.* Layout of the physics-informed *prescribed* CSGMM.

## 4.1. Incorporating Physics via Implicit Modeling

One possibility to address this problem is by utilizing the *implicit* AmbientGAN described in Section 3.3. By combining (9) and the compressible representation of channels, cf. (5), we identify the mapping $\boldsymbol{f}(\cdot)$ of AmbientGANs by

$$\boldsymbol{y}_i = \boldsymbol{f}(\boldsymbol{s}_i) = \boldsymbol{A}\boldsymbol{D}\boldsymbol{s}_i + \boldsymbol{n}_i \qquad (10)$$

with $\boldsymbol{n}_i$ being AWGN exhibiting the same statistical characteristics as the noise present in the training dataset. A schematic of the resulting AmbientGAN is given in Fig. 2. For one forward operation during training, we first generate several $\boldsymbol{z}_i$ drawn from a simplistic distribution (e.g., $\mathcal{N}(\boldsymbol{0}, \boldsymbol{I})$). These samples are then forwarded by the generator $\mathcal{G}_{\boldsymbol{\phi}}(\cdot)$ and the mapping $\boldsymbol{f}(\cdot)$, which in turn is characterized by the pre-defined dictionary $\boldsymbol{D}$, pre-known measurement matrix $\boldsymbol{A}$ and noise $\boldsymbol{n}_i$. This procedure results in the generated $\boldsymbol{\phi}$-differentiable observations $\boldsymbol{y}_{\text{gen},i}$

$$\boldsymbol{y}_{\text{gen},i} = \boldsymbol{A}\boldsymbol{D}\mathcal{G}_{\boldsymbol{\phi}}(\boldsymbol{z}_i) + \boldsymbol{n}_i. \qquad (11)$$

In the final step of one forward operation, the discriminator $\mathcal{D}_{\boldsymbol{\theta}}(\cdot)$ inputs $\boldsymbol{y}_{\text{gen},i}$ as well as training samples $\boldsymbol{y}_{\text{real},i} \in \mathcal{Y}$ and approximates a $\boldsymbol{\theta}$-differentiable quantity measuring the distance between the desired and currently learned distribution. The NN weights $(\boldsymbol{\phi}, \boldsymbol{\theta})$ are learned by solving a minimax optimization problem with Wasserstein distance-based objective (Gulrajani et al., 2017). A detailed explanation is given in Appendix A.2. After training, a new $\boldsymbol{s}$ can be generated by drawing a $\boldsymbol{z}$ and computing $\mathcal{G}_{\boldsymbol{\phi}}(\boldsymbol{z})$.

## 4.2. Incorporating Physics via Prescribed Modeling

Building on *prescribed* generative modeling, we can also use SBGM, and in particular the CSVAE and CSGMM, explained in Section 3.3 to learn $p(\boldsymbol{s})$ from $\mathcal{Y}$.

Fig. 3 shows a schematic of the CSVAE when being applied to learn $p(\boldsymbol{s})$. Equivalent to VAEs, the training of the CSVAE is based on a lower bound of the log-evidence (Böck et al., 2024b), i.e.,

$$\sum_{\boldsymbol{y}_i \in \mathcal{Y}} \Big( \mathbb{E}_{p_{\boldsymbol{\theta}}(\boldsymbol{s}|\tilde{\boldsymbol{z}}_i, \boldsymbol{y}_i)}[\log p(\boldsymbol{y}_i|\boldsymbol{s})] - \qquad (12)$$
$$\mathrm{D}_{\mathrm{KL}}(q_{\boldsymbol{\phi}}(\boldsymbol{z}|\boldsymbol{y}_i)\|p(\boldsymbol{z})) - \mathrm{D}_{\mathrm{KL}}(p_{\boldsymbol{\theta}}(\boldsymbol{s}|\tilde{\boldsymbol{z}}_i, \boldsymbol{y}_i)\|p_{\boldsymbol{\theta}}(\boldsymbol{s}|\tilde{\boldsymbol{z}}_i)) \Big)$$

with $\tilde{\boldsymbol{z}}_i$ drawn from a variational distribution $q_{\boldsymbol{\phi}}(\boldsymbol{z}|\boldsymbol{y}_i) = \mathcal{N}(\boldsymbol{z}; \boldsymbol{\mu}_{\boldsymbol{\phi}}(\boldsymbol{y}), \mathrm{diag}(\boldsymbol{\sigma}_{\boldsymbol{\phi}}^2(\boldsymbol{y})))$. The distribution $p_{\boldsymbol{\delta}}(\boldsymbol{z})$ in (8)

is assumed to be $\mathcal{N}(\boldsymbol{0}, \boldsymbol{I})$. One forward operation during training consists of first inputing training samples $\boldsymbol{y}_i$ to the encoder (Enc) to yield $\boldsymbol{\mu}_{\boldsymbol{\phi}}(\boldsymbol{y}_i)$ and $\boldsymbol{\sigma}_{\boldsymbol{\phi}}(\boldsymbol{y}_i)$. We then forward the samples $\tilde{\boldsymbol{z}}_i$ drawn from $q_{\boldsymbol{\phi}}(\boldsymbol{z}|\boldsymbol{y}_i)$ to the decoder (Dec) whose output represents $\boldsymbol{\gamma}_{\boldsymbol{\theta}}(\tilde{\boldsymbol{z}}_i)$ fully characterizing $p_{\boldsymbol{\theta}}(\boldsymbol{s}|\tilde{\boldsymbol{z}}_i)$ (cf. (7)). A key aspect in SBGM is that $p_{\boldsymbol{\theta}}(\boldsymbol{s}|\tilde{\boldsymbol{z}}_i, \boldsymbol{y}_i)$ is a Gaussian with closed-form mean and covariance specified in Appendix A.3. Based on these moments and $(\boldsymbol{\mu}_{\boldsymbol{\phi}}(\boldsymbol{y}_i), \boldsymbol{\sigma}_{\boldsymbol{\phi}}(\boldsymbol{y}_i), \boldsymbol{\gamma}_{\boldsymbol{\theta}}(\tilde{\boldsymbol{z}}_i))$ derived during the forward operation, all terms in (12) can be calculated and differentiated with respect to $(\boldsymbol{\theta}, \boldsymbol{\phi})$. The closed forms and a more detailed explanation are given in Appendix A.4. Note that we solely require the processing chain in Fig. 3 up to $\sqrt{\boldsymbol{\gamma}_{\boldsymbol{\theta}}}$ for training. After training, we generate a new sample $\boldsymbol{s}$ by first drawing a $\boldsymbol{z} \sim p_{\boldsymbol{\delta}}(\boldsymbol{z})$ and forwarding it through the decoder. This generates $\sqrt{\boldsymbol{\gamma}_{\boldsymbol{\theta}}(\boldsymbol{z})}$, which allows us to draw a sample $\boldsymbol{s} \sim p_{\boldsymbol{\theta}}(\boldsymbol{s}|\boldsymbol{z})$ (cf. (7)). The necessity of this second sampling operation marks the key difference between the *prescribed* and *implicit* modeling approach.[4]

Similar to the schematic in Fig. 3, Fig. 4 shows a schematic of the CSGMM when being applied to learn $p(\boldsymbol{s})$. For the CSGMM, the distribution $p_{\boldsymbol{\delta}}(\boldsymbol{z})$ is a categorial distribution, i.e., $p_{\boldsymbol{\delta}}(\boldsymbol{z}) = p_{\boldsymbol{\delta}}(k) = \rho_k$ $(k = 1, \dots, K)$. Moreover, $\boldsymbol{\gamma}_{\boldsymbol{\theta}}(\boldsymbol{z}) = \boldsymbol{\gamma}_k$, i.e., $\boldsymbol{s}$ follows a GMM with zero means and diagonal covariance matrices. The learning of $\{\rho_k, \boldsymbol{\gamma}_k\}_{k=1}^K$ is done by an extended expectation-maximization (EM) algorithm considering the additional latent variable $\boldsymbol{s}$ in (6)-(8) (Böck et al., 2024b). In the E-step, the model's posterior $p(\boldsymbol{s}, k|\boldsymbol{y}_i) = p(\boldsymbol{s}|k, \boldsymbol{y}_i)p(k|\boldsymbol{y}_i)$ has to be computed for each training sample $\boldsymbol{y}_i$. Equivalent to the CSVAE, $p(\boldsymbol{s}|k, \boldsymbol{y}_i)$ is Gaussian with closed-form mean and covariance (cf. Appendix A.3) also resulting in a closed-form E-step. The M-step equals the one in (Böck et al., 2024b). Both are summarized in Appendix A.5. Similar to CSVAEs, we generate a new sample $\boldsymbol{s}$ after training by first drawing a $k \sim p_{\boldsymbol{\delta}}(k)$ and then another sample $\boldsymbol{s} \sim p_{\boldsymbol{\theta}}(\boldsymbol{s}|k)$ (cf. (7)).

---

[4]Note that VAEs can also be used as *implicit* models by having the decoder output only means, cf. (Dai & Wipf, 2019).

## 4.3. Why SBGM Is Capable of Parameter Generation

In (Böck et al., 2024b), SBGM is introduced to directly solve inverse problems, and not to generate new samples. Indeed, (Böck et al., 2024b) discusses that SBGM is not suited for generation due to its restriction on $s$ to have a universal conditional zero mean, i.e., $\mathbb{E}[s|z] = 0$ for all $z$ (cf. (7)). This assumption strictly limits its capability of generating realistic samples as it is typically not possible to decompose the true unknown distribution $p(s)$ in this way (Böck et al., 2024b). This raises the question whether this limitation also hinders SBGM from properly generating wireless channels. However, as the following discussion shows, wireless channels exhibit a unique property that eliminates this limitation in its entirety when incorporating the common assumption of stationarity. Moreover, we show that the conditional zero mean and diagonal covariance property in (7) perfectly aligns with model-based insights about the structure of conditional channel moments in general.

The work in (Böck et al., 2024a) explores whether the structural properties of the first and second channel moments (i.e., $\mathbb{E}[h]$ and $\mathbb{E}[hh^{\mathrm{H}}]$) remain intact when conditioning the channel on some side information $z$. This publication considers the channel representation which is based on the sum of steering vectors as it is done in (2) and (3). In particular, it shows that under some mild conditions, such as stationarity, the channel is guaranteed to have a conditional zero mean and (block-) Toeplitz-structured covariance, i.e.,[5]

$$\mathbb{E}[h|z] = 0, \mathbb{E}[hh^{\mathrm{H}}|z] \text{ (block-)Toeplitz} \qquad (13)$$

if $z$ does not contain information about the phases of the path losses $\rho_{\ell,m}$ in (1). Noting that these phases cannot represent site-specific channel features, (Böck et al., 2024a) argues and empirically demonstrates that the latent variable $z$ in CGLMs satisfies this requirement. As a result, when restricting any CGLM (e.g., VAEs or GMMs) to satisfy (13), we enforce the model to be physically consistent.

SBGM models $s$ such that $\mathbb{E}[s|z] = 0$ and $\mathbb{E}[ss^{\mathrm{H}}|z]$ to be diagonal (cf. (7)). Moreover, the channel $h$ and $s$ are related linearly by a dictionary $D$ (cf. (5)). Thus, SBGM enforces the channel $h$ to exhibit

$$\mathbb{E}[h|z] = D\,\mathbb{E}[s|z] = 0, \mathbb{E}[hh^{\mathrm{H}}|z] = D\mathrm{diag}(\gamma_{\theta}(z))D^{\mathrm{H}}. \qquad (14)$$

Latter is (block-)Toeplitz when inserting the dictionaries from Section 3.2, and, thus, SBGM perfectly aligns with (13). In consequence, the conditional zero mean and diagonal covariance property in SBGM (cf. (7)) is no limitation for generating wireless channels, but rather the opposite. It regularizes the search space of the statistical models to those that are physically consistent.[6]

---

[5]A block of Toeplitz matrices arises when considering multiple domains at once, e.g., time and frequency in OFDM.

[6]A more detailed explanation is given in Appendix A.6.

## 5. Discussion

**Generalizability**   Section 4 addresses how the physics-informed generative approaches circumvent the need for high-quality training data and provide physical interpretability. One more claim from Section 1 is generalizability, i.e., the adaptability to other system configurations, such as different numbers of antennas or subcarrier spacings after training. The link between the compressible channel representation $s$ and the physical parameters $\mathcal{P}$ is solely determined by the choice of the parameter grid $\mathcal{G}$ (cf. Section 3.2). On the other hand, the system configuration is solely encoded in the columns of the dictionary $D$, but the parameter grid $\mathcal{G}$ is system configuration-independent. Thus, the learned channel parameter generators in Section 4 (cf. Fig. 2-4) do not depend on the dictionary used for training. In consequence, when sampling a new $s$ from the generative model after training and computing the corresponding channel realization $h$, we can use a new dictionary $D^{(\mathrm{new})}$, whose domain must match the dictionary used for training, but whose range can be easily adapted to a newly desired system configuration without any retraining.

**Providing Training Data for Wireless Communication**
Most ML-based methods for the physical layer in wireless communications either require lots of ground-truth and site-specific channel realizations for training or at least site-specific pairs of input and output signals linked by the wireless channels. However, finding efficient and scalable ways to obtain this data in each scenario of interest is an open challenge (Kim et al., 2023). With the models from Section 4, we present a solution to this problem. These methods can learn from corrupted data that a wireless receiver obtains, e.g., during default online operation. It then can generate a desired amount of site-specific clean channel realizations for, e.g., training ML-based methods.

**Implicit versus Prescribed Modeling**   The *implicit* and *prescribed* modeling approaches in Section 4 share the properties of not requiring ground-truth data for training, being physically interpretable, and generalizable over system configurations. However, there are also two key properties in which these approaches differ from each other, rendering the *prescribed* approach to be generally superior. As the compressible channel representation $s$ typically exhibits a much larger dimension than the channel observation $y$, learning $p(s)$ from a dataset $\mathcal{Y} = \{y_i\}_{i=1}^{N_t}$ is an ill-posed problem. Therefore, to properly learn $p(s)$, some regularization is required. On the one hand, when applying SBGM for learning $p(s)$, we impose the general sparsity-promoting regularization of SBGM in $s$ (cf. Section 3.3). On the other hand, as discussed in Section 4.3, this regularization extends beyond sparsity and additionally enforces the learning process to yield a physically consistent model. In comparison, the *im-*

*Table 1.* Comparison between different channel modeling schemes.

| Channel Model | Requires **No** High Quality Training Data/ Measurement Campaigns | Site-Specific | **No** Accuracy-Simplicity Trade-Off | Generalizable to other System Configurations | Physically Interpretable | Guaranteed Physical Consistency | Sparsity-Inducing |
|---|---|---|---|---|---|---|---|
| Standardized Channel Model (e.g., 3GPP)) | | | | ✓ | ✓ | ✓ | ✓ |
| Black-Box Gen. Modeling (Learning From $h$ and Generating $h$) | ✓ | | ✓ | | | | |
| Physics-Informed *Implicit* Gen. Modeling (Section 4.1) | ✓ | ✓ | ✓ | ✓ | ✓ | | |
| *Prescribed* SBGM (e.g., CSVAE, CSGMM) (Section 4.2) | ✓ | ✓ | ✓ | ✓ | ✓ | ✓ | ✓ |

*plicit*-based learning process in Section 4.1 allows to learn $p(s)$ from $\mathcal{Y}$, but imposes neither sparsity nor physical consistence during training and, thus, keeps the overall problem ill-posed. Table 1 provides a comprehensive comparison between the *implicit* and *prescribed* modeling approaches as well as standardized channel models and black-box-based generative modeling, both discussed in Section 1.

**Foundation Modeling Perspective**  While SBGM can generate training data, it can also be directly used for real-time processing tasks. Recently, VAEs and GMMs have been used for channel estimation (Baur et al., 2024a; Böck et al., 2023; Fesl et al., 2024; Koller et al., 2022), channel prediction (Turan et al., 2024a), and precoder- and pilot design (Turan et al., 2024b; 2025). Thus, the CSVAE and CSGMM not only generalize to arbitrary system configurations but also provide information that can be directly used for these downstream tasks without the need for specific training. This aligns with the idea of foundation modeling, i.e., training a generalized ML model and applying it to various downstream tasks without retraining.

**Limitations**  Both proposed approaches in Section 4 have limitations related to non-stationary channel generation, off-grid mismatches, and pilot patterns, cf. Appendix B.

**Model Extensions**  Using ray tracing for wireless channel modeling allows customizing the number of paths for each channel (Alkhateeb, 2019). SBGM offers the same flexibility, cf. Appendix C.1. Moreover, when considering multiple domains at once (e.g., OFDM), one can reduce the number of learnable parameters by further constraining the learned variances in SBGM, as detailed in Appendix C.2.

## 6. Experiments

### 6.1. Experimental Setup

**Datasets**  For evaluation, we use four datasets. We use a modified standardized 3GPP spatial channel model for SIMO, which we adapted to better illustrate our method. For simulations with OFDM, we use two different QuaDRiGa-based datasets (Jaeckel et al., 2014). One (5G-Urban)

represents an urban macro-cell, in which users can be in line-of-sight (LOS), non-LOS (NLOS), as well as indoor and outdoor. The other (5G-Rural) represents a rural macro-cell, in which all users are in LOS. We also use the ray tracing database DeepMIMO (Alkhateeb, 2019) for SIMO in Appendix F. For the channel observations in OFDM, we generate one random selection matrix $A$ extracting $M$ entries from $h$ and apply it to every training channel (cf. (9)). In all simulations and each training sample, we draw signal-to-noise ratios (SNRs) uniformly distributed between 5 and 20dB, defining the noise variance $\sigma_i^2$ (cf. Section 4). A detailed description of the datasets, chosen configurations, and pre-processing is given in Appendix D.

**Evaluation metrics**  We evaluate the parameter generation performance by the power angular profile

$$P_\omega^{(\mathrm{R})}(q) = \frac{1}{N_{\text{test}}} \sum_{i=1}^{N_{\text{test}}} \frac{|s_{\mathrm{R},i}^{(q)}|^2}{\sum_{g=-S_{\mathrm{R}}/2}^{S_{\mathrm{R}}/2-1} |s_{\mathrm{R},i}^{(g)}|^2} \quad (15)$$

as well as the channel-wise angular spread

$$S_\omega^{(\mathrm{R})}(s_{\mathrm{R}}) = \sqrt{\frac{\sum_{g=-S_{\mathrm{R}}/2}^{S_{\mathrm{R}}/2-1} (\omega_g^{(\mathrm{R})} - \mu_{\mathrm{R}}^{(g)})^2 |s_{\mathrm{R}}^{(g)}|^2}{\sum_{g=-S_{\mathrm{R}}/2}^{S_{\mathrm{R}}/2-1} |s_{\mathrm{R}}^{(g)}|^2}} \quad (16)$$

with $s_{\mathrm{R},i}^{(g)}$ being the $g$th entry in the $i$th newly generated sample $s_{\mathrm{R},i}$. Moreover, $\omega_g^{(\mathrm{R})} = g\pi/S_{\mathrm{R}}$ and $\mu_{\mathrm{R},i}^{(g)} = (\sum_{g=-S_{\mathrm{R}}/2}^{S_{\mathrm{R}}/2-1} \omega_g^{(\mathrm{R})}|s_{\mathrm{R},i}^{(g)}|^2)/(\sum_{g=-S_{\mathrm{R}}/2}^{S_{\mathrm{R}}/2-1} |s_{\mathrm{R},i}^{(g)}|^2)$ (Zhang et al., 2017). For the channel generation performance, we map newly generated $s_{\mathrm{R}}$ (or $s_{t,f}$) to $h$ using a dictionary $D_{\mathrm{R}}$ (or $D_{t,f}$) (cf. Section 3.2) and evaluate the channel generation with the cross-validation method from (Baur et al., 2025; Xiao et al., 2022). Specifically, we first train each generative model using $\mathcal{Y}$ (cf. Section 4) and generate $N_{\text{(gen)}}$ channels with each model to train an autoencoder for reconstruction by minimizing the mean squared error (MSE) for each generative model separately. We then compress and reconstruct ground-truth (i.e., QuaDRiGa) channels using these trained autoencoders and evaluate the normalized MSE (NMSE) $\text{nMSE} = 1/N_{\text{test}} \sum_{i=1}^{N_{\text{test}}} (\|\hat{h}_i - h_i\|_2^2/N)$ and the cosine similarity $\rho_c = 1/N_{\text{test}} \sum_{i=1}^{N_{\text{test}}} (|\hat{h}_i^{\mathrm{H}} h_i|/(\|\hat{h}_i\|_2 \|h_i\|_2))$ with $h_i$ and $\hat{h}_i$ being the ground-truth and reconstructed channel.

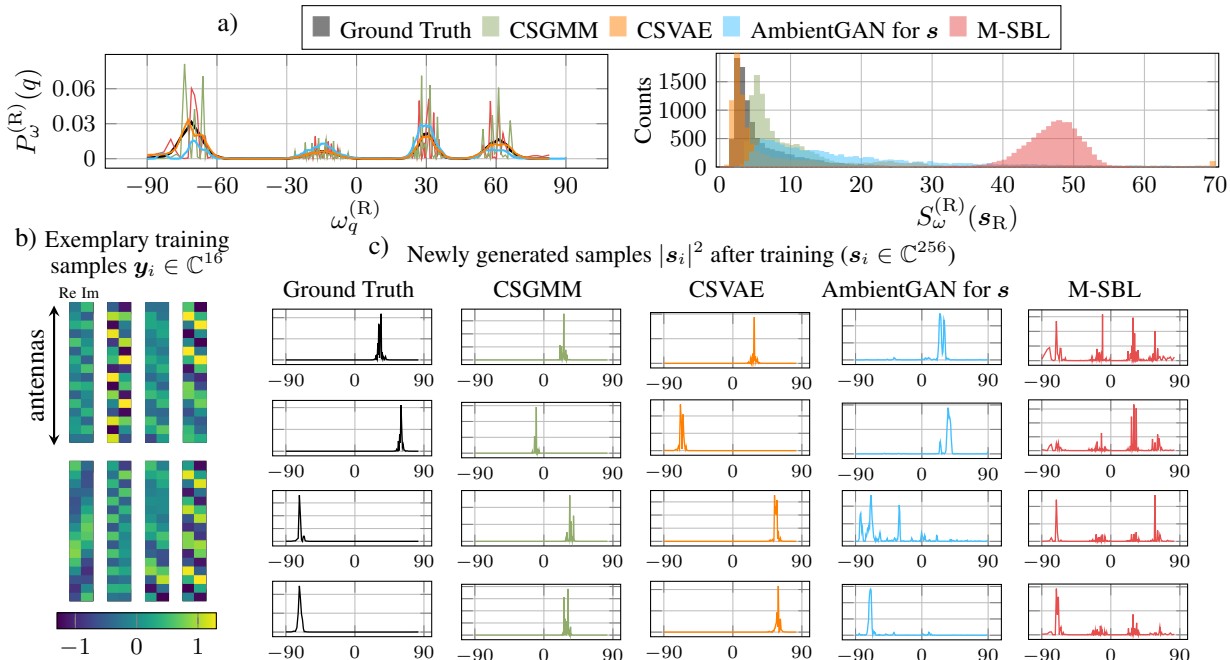

*Figure 5.* a) Power angular profile $P_\omega^{(R)}(q)$ and a histogram of the angular spread $S_\omega^{(R)}(s_R)$ from 10 000 generated samples by CSVAE, CSGMM, AmbientGAN for $s$ and M-SBL, compared to ground truth, b) eight exemplary training samples, c) squared absolute value of four exemplary generated samples from all models, respectively.

**Baselines & architectures** For parameter generation in SIMO, we compare the *prescribed* CSVAE, the *prescribed* CSGMM (cf. Section 4.2), the *implicit* AmbientGAN trained to output $s$ (cf. Section 4.1), and M-SBL (Wipf & Rao, 2007). While M-SBL was not originally designed for generation, it is equivalent to the CSGMM from (Böck et al., 2024b) with $K = 1$ component, effectively fitting a Gaussian to $s$ that can be used for sampling. For generating channel realizations, we also evaluate an AmbientGAN (Bora et al., 2018) that learns to directly output $h$. A detailed overview of all architectures, hyperparameters, and the ground-truth baseline is given in Appendix E.

### 6.2. Results

**Modified 3GPP** In Fig. 5 a), the power angular profile $P_\omega^{(R)}(q)$ as well as a histogram of the angular spread $S_\omega^{(R)}(s_R)$ is given. The number of antennas $N = M$ is set to 16, and the number of gridpoints $S$ is set to 256. The number $N_t$ of training samples is 10 000. In Fig. 5 b) and c), exemplary training samples and newly generated samples are shown. In general, all power angular profiles are consistent with ground truth by, e.g., not assigning power to directions absent in the ground-truth profile. CSGMM and M-SBL produce sharper power angular profiles with more peaks than the ground truth. CSVAE yields a power angular profile that best matches the ground truth. When considering the histogram of angular spreads, the methods exhibit very different characteristics. While CSGMM slightly overestimates the angular spread, and CSVAE exhibits a few

outliers with a large angular spread, both closely resemble ground truth. In contrast, the *implicit* AmbientGAN significantly overestimates the angular spread. This is due to the missing promotion of sparsity and guarantee of physical consistency during training (cf. Section 5). Since M-SBL fits a Gaussian to $s$, it can not assign different directions to different samples, resulting in largely overestimating the angular spread. Since M-SBL equals CSGMM with $K = 1$ component, this histogram also illustrates the advantage of CSGMM having more than one component. The different characteristics are also illustrated by the generated samples in Fig. 5 c). CSGMM and CSVAE generate samples that resemble ground truth, whereas AmbientGAN produces samples with a broader angular spread, and M-SBL encodes all directions of the power angular profile into every sample.

**QuaDRiGa** In Fig. 6 a) and b), the nMSE and $\rho_c$ of the autoencoder reconstruction are shown over the number $N_t$ of training samples with fixed $M = 30$ for the QuaDRiGa dataset `5G-Urban` (cf. Appendix D.2). We choose $S_t = S_f = 40$, $\bar\tau = 6\mu s$ and $\bar\vartheta = 0.25$kHz (cf. Section 3.2). In all simulations, $N_{(gen)} = 30 000$. AmbientGAN for $h$ performs significantly worse than all other methods. AmbientGAN for $s$ improves over $N_t$, but does not outperform M-SBL. Overall, CSVAE and CSGMM achieve the best results. Notably, CSGMM closely approaches ground truth performance even for small $N_t$, for which we train the autoencoder by means of 30 000 ground-truth channels. Given that M-SBL fits a Gaussian to $s$ and performs consistently well, we infer that the true $p(s)$ in the `5G-Urban`

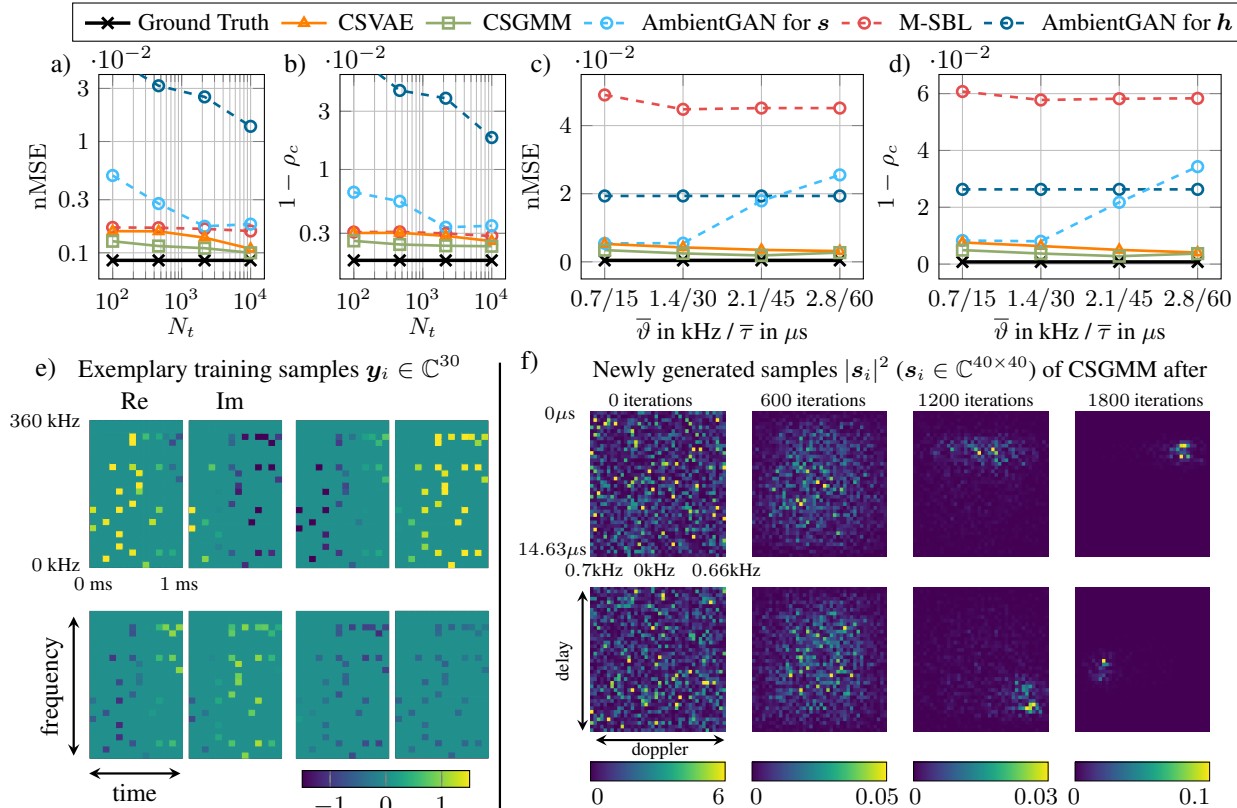

*Figure 6.* a) - d) nMSE and $\rho_c$ for reconstructing ground-truth channels by an autoencoder trained on channels $\boldsymbol{h} = \boldsymbol{D}\boldsymbol{s}$ produced by the trained models, respectively. In a) and b), we vary $N_t$ for the generative models with fixed $M = 30$ and dataset `5G-Urban`, and in c) and d), we vary $\bar{\vartheta}$ and $\bar{\tau}$ with fixed $N_t = 3\,000$ and dataset `5G-Rural`, e) four exemplary training samples, f) illustration of the CSGMM training by the squared absolute value of two generated samples after 0, 600, 12000 and 1800 iterations.

dataset exhibits Gaussian characteristics. This is in contrast to the results in Fig. 6 c) and d), where we used the `5G-Rural` dataset (cf. Appendix D.2). Here, we set $N_t = 3\,000$, but vary the maximally resolvable Doppler $\bar{\vartheta}$ and delay $\bar{\tau}$. We choose $S_t = S_f = 40$, $M = 30$. Overall, M-SBL performs the worst. In general, $\bar{\vartheta}$ and $\bar{\tau}$ are regularizing hyperparameters. When choosing both to just encompass all occurring ground-truth delays and Dopplers without causing ambiguities, the *implicit* AmbientGAN shows good performance. However, increasing $\bar{\vartheta}$ and $\bar{\tau}$ weakens this regularization, leading to a significant drop in AmbientGAN's performance. CSGMM and CSVAE additionally regularize their training by inducing sparsity and ensuring physical consistency. Thus, they maintain strong performance even when $\bar{\vartheta}$ and $\bar{\tau}$ are large. Fig. 6 e) shows four exemplary training samples. To illustrate the OFDM masking with pilot symbols, we plot the real (Re) and imaginary (Im) part of the 30-dimensional observations within the OFDM grid representing the underlying channel $\boldsymbol{H} \in \mathbb{C}^{24 \times 14}$ with $\boldsymbol{h} = \text{vec}(\boldsymbol{H})$. Fig. 6 f) illustrates the training of CSGMM when being trained on `5G-Rural`. Specifically, we plot the squared absolute value of two samples in the delay-Doppler domain after 0, 600, 1200, and 1800 training iterations, demonstrating the increase in sparsity during training.

**Additional Results** In Appendix F.1, we analyze the generalizability of SBGM to other system configurations (cf. Section 5). Moreover, in Appendix F.2 and F.3, we evaluate controlling the number of paths per sample, and the reduction of learnable parameters by enforcing additional structure. In Appendix F.4, we also test the parameter generation for the DeepMIMO dataset. In Appendix G, we provide pseudocode and some implementation specifications.

## 7. Conclusion

In this work, we combined the physics-related compressibility of wireless channels with *prescribed* and *implicit* generative models to yield a physics-informed generative modeling framework for wireless channels. We also established that due to the promotion of sparsity and guarantee of physical consistency, SBGM as *prescribed* model is superior to *implicit* alternatives. These methods can generate the physical channel parameters and channel realizations themselves. We validated that the parameter generation is consistent with ground truth and that the introduced models outperform GAN-based black-box baselines for generating channel realizations. Limitations, such as generating non-stationary channel trajectories, are part of future work.

## Impact Statement

This paper presents work whose goal is to advance the field of Machine Learning. There are many potential societal consequences of our work, none which we feel must be specifically highlighted here.

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

## A. Additional Explanations

### A.1. Channel Sparsity in OFDM Systems

The OFDM channel matrix is given by (3), i.e.,

$$\boldsymbol{H} = \sum_{\ell=1}^{L} \sum_{m=1}^{M_\ell} \rho_{\ell,m} \boldsymbol{a}_t(\vartheta_{\ell,m}) \boldsymbol{a}_f(\tau_{\ell,m})^{\mathrm{T}} \tag{17}$$

with $\boldsymbol{a}_t(\vartheta_{\ell,m})\big|_i = \mathrm{e}^{\mathrm{j}\,2\pi\vartheta_{\ell,m}(i-1)\Delta T}$ and $\boldsymbol{a}_f(\tau_{\ell,m})\big|_j = \mathrm{e}^{-\mathrm{j}\,2\pi\tau_{\ell,m}(j-1)\Delta f}$.[7] We assume that we have access to (not necessarily tight) upper bounds $\overline{\tau}$ and $\overline{\vartheta}$ for all delays and doppler shifts, i.e.,

$$\max_{\ell,m} \tau_{\ell,m} < \overline{\tau} \tag{18}$$

$$\max_{\ell,m} |\vartheta_{\ell,m}| \leq \overline{\vartheta} \tag{19}$$

By deciding for a number of grid points $S_f$ and $S_t$ in the delay and doppler domain, respectively, we define the grids $\mathcal{G}_f = \{j\overline{\tau}/S_f\}_{j=0}^{S_f-1}$ as well as $\mathcal{G}_t = \{i2\overline{\vartheta}/S_t\}_{i=-S_t/2}^{S_t/2-1}$. In consequence, we can represent the OFDM channel matrix in (17) using $\mathcal{G}_f \times \mathcal{G}_t$ as

$$\boldsymbol{h} = \sum_{i=-S_t/2}^{S_t/2-1} \sum_{j=0}^{S_f-1} s_{t,f}^{(i,j)} \mathrm{vec}\left( \boldsymbol{a}_t(\frac{i2\overline{\vartheta}}{S_t}) \boldsymbol{a}_f(\frac{j\overline{\tau}}{S_f})^{\mathrm{T}} \right) = \boldsymbol{D}_{t,f} \boldsymbol{s}_{t,f}, \tag{20}$$

with $\boldsymbol{s}_{t,f} = \mathrm{vec}(\boldsymbol{S}_{t,f})$, $\boldsymbol{S}_{t,f}\big|_{i,j} = s_{t,f}^{(i,j)}$ and $\boldsymbol{D}_{t,f} = \boldsymbol{D}_t \otimes \boldsymbol{D}_f$. Moreover,

$$\boldsymbol{D}_t = [\boldsymbol{a}_t(-\overline{\vartheta}), \ldots, \boldsymbol{a}_t(\overline{\vartheta} - \frac{2\overline{\vartheta}}{S_t})] \tag{21}$$

$$\boldsymbol{D}_f = [\boldsymbol{a}_f(0), \ldots, \boldsymbol{a}_f(\overline{\tau} - \frac{\overline{\tau}}{S_f})] \tag{22}$$

and we know $\boldsymbol{s}_{t,f}$ to be compressible in general.

### A.2. Detailed Explanation of Training the *Implicit* AmbientGAN

The implementation of the *implicit* AmbientGAN is based on the Wasserstein GAN with gradient penalty (Gulrajani et al., 2017). In particular, the min-max objective function is given by

$$\min_{\boldsymbol{\phi}} \max_{\boldsymbol{\theta}} \mathbb{E}[\mathcal{D}_{\boldsymbol{\theta}}(\boldsymbol{A}\boldsymbol{D}\mathcal{G}_{\boldsymbol{\phi}}(\boldsymbol{z}) + \boldsymbol{n})] - \mathbb{E}[\mathcal{D}_{\boldsymbol{\theta}}(\boldsymbol{y})] + \lambda\, \mathbb{E}[(\|\nabla \mathcal{D}_{\boldsymbol{\theta}}(\hat{\boldsymbol{y}})\|_2 - 1)^2] \tag{23}$$

with $\hat{\boldsymbol{y}} = \epsilon \boldsymbol{y} + (1-\epsilon)(\boldsymbol{A}\boldsymbol{D}\mathcal{G}_{\boldsymbol{\phi}}(\boldsymbol{z}) + \boldsymbol{n})$ and $\epsilon \sim \mathcal{U}(0, 1)$. The inner maximization realizes the maximization included in the Wasserstein distance according to the Kantorovich-Rubinstein duality, and with the outer minimization, we aim to minimize this distance. The third term regularizes $\mathcal{D}_{\boldsymbol{\theta}}(\cdot)$ towards being Lipschitz-1 continuous, required for the proper definition of the Wasserstein distance (Arjovsky et al., 2017). The second expectation can be estimated batch-wise using the training dataset, while the first one can be estimated by Monte-Carlo sampling $\boldsymbol{z}$ and $\boldsymbol{n}$ from their known distributions $p(\boldsymbol{z})$ and $p(\boldsymbol{n})$. The parameter $\lambda$ is a hyperparameter.

### A.3. Detailed Description of the Closed-Form Moments of $p_{\boldsymbol{\theta}}(\boldsymbol{s}|\boldsymbol{z}, \boldsymbol{y})$

The distribution $p_{\boldsymbol{\theta}}(\boldsymbol{s}|\boldsymbol{z}, \boldsymbol{y})$ forms a conditioned conjugate prior of $\boldsymbol{y}|\boldsymbol{s}$ (conditioned on $\boldsymbol{z}$). Thus, we can compute the moments of $p_{\boldsymbol{\theta}}(\boldsymbol{s}|\tilde{\boldsymbol{z}}_i, \boldsymbol{y}_i)$ as (see Appendix I in (Böck et al., 2024b))

$$\boldsymbol{\mu}_{\boldsymbol{\theta}}^{\boldsymbol{s}|\boldsymbol{y}_i,\tilde{\boldsymbol{z}}_i}(\tilde{\boldsymbol{z}}_i) = \boldsymbol{C}_{\boldsymbol{\theta}}^{\boldsymbol{s},\boldsymbol{y}_i|\tilde{\boldsymbol{z}}_i}(\tilde{\boldsymbol{z}}_i) \left( \boldsymbol{C}_{\boldsymbol{\theta}}^{\boldsymbol{y}_i|\tilde{\boldsymbol{z}}_i}(\tilde{\boldsymbol{z}}_i) \right)^{-1} \boldsymbol{y}_i \tag{24}$$

$$\boldsymbol{C}_{\boldsymbol{\theta}}^{\boldsymbol{s}|\boldsymbol{y}_i,\tilde{\boldsymbol{z}}_i}(\tilde{\boldsymbol{z}}_i) = \mathrm{diag}(\boldsymbol{\gamma}_{\boldsymbol{\theta}}(\tilde{\boldsymbol{z}}_i)) - \boldsymbol{C}_{\boldsymbol{\theta}}^{\boldsymbol{s},\boldsymbol{y}_i|\tilde{\boldsymbol{z}}_i}(\tilde{\boldsymbol{z}}_i) \left( \boldsymbol{C}_{\boldsymbol{\theta}}^{\boldsymbol{y}_i|\boldsymbol{z}}(\tilde{\boldsymbol{z}}_i) \right)^{-1} \boldsymbol{C}_{\boldsymbol{\theta}}^{\boldsymbol{s},\boldsymbol{y}_i|\tilde{\boldsymbol{z}}_i}(\tilde{\boldsymbol{z}}_i)^{\mathrm{H}} \tag{25}$$

---

[7]The term $\boldsymbol{a}_{\mathrm{R}}(\boldsymbol{\Omega}_{\ell,m}^{(\mathrm{R})}) \boldsymbol{a}_{\mathrm{T}}(\boldsymbol{\Omega}_{\ell,m}^{(\mathrm{T})})^{\mathrm{T}}$ is also sometimes represented by its equivalent tensor product $\boldsymbol{a}_{\mathrm{R}}(\boldsymbol{\Omega}_{\ell,m}^{(\mathrm{R})}) \otimes \boldsymbol{a}_{\mathrm{T}}(\boldsymbol{\Omega}_{\ell,m}^{(\mathrm{T})})$, cf. (Böck et al., 2024a).

with

$$C_{\boldsymbol{\theta}}^{\boldsymbol{y}_i|\tilde{\boldsymbol{z}}_i}(\tilde{\boldsymbol{z}}_i) = \boldsymbol{A}\boldsymbol{D}\text{diag}(\boldsymbol{\gamma}_{\boldsymbol{\theta}}(\tilde{\boldsymbol{z}}_i))\boldsymbol{D}^{\text{H}}\boldsymbol{A}^{\text{H}} + \sigma_i^2\,\mathbf{I},\tag{26}$$

and $C_{\boldsymbol{\theta}}^{\boldsymbol{s},\boldsymbol{y}_i|\tilde{\boldsymbol{z}}_i}(\tilde{\boldsymbol{z}}_i) = \text{diag}(\boldsymbol{\gamma}_{\boldsymbol{\theta}}(\tilde{\boldsymbol{z}}_i))\boldsymbol{D}^{\text{H}}\boldsymbol{A}^{\text{H}}$. Minor differences to the setup in (Böck et al., 2024b) are the sample-dependent noise variance $\sigma_i^2$ in (26) as well as taking the Hermitian instead of the transpose.

One efficient and differentiable implementation of computing the inverse in (24) and (25) is by first calculating the Cholesky decomposition $\boldsymbol{L}_{\boldsymbol{\theta}}^{\boldsymbol{y}_i|\tilde{\boldsymbol{z}}_i}(\tilde{\boldsymbol{z}}_i)\boldsymbol{L}_{\boldsymbol{\theta}}^{\boldsymbol{y}_i|\tilde{\boldsymbol{z}}_i}(\tilde{\boldsymbol{z}}_i)^{\text{H}}$ of $\boldsymbol{C}_{\boldsymbol{\theta}}^{\boldsymbol{y}_i|\tilde{\boldsymbol{z}}_i}(\tilde{\boldsymbol{z}}_i)$, then computing $\boldsymbol{L}_{\boldsymbol{\theta}}^{\boldsymbol{y}_i|\tilde{\boldsymbol{z}}_i}(\tilde{\boldsymbol{z}}_i)^{-1}$ by solving $M$ linear systems of equations with triangular matrix $\boldsymbol{L}_{\boldsymbol{\theta}}^{\boldsymbol{y}_i|\tilde{\boldsymbol{z}}_i}(\tilde{\boldsymbol{z}}_i)$, and finally computing $\boldsymbol{L}_{\boldsymbol{\theta}}^{\boldsymbol{y}_i|\tilde{\boldsymbol{z}}_i}(\tilde{\boldsymbol{z}}_i)^{-1}\boldsymbol{L}_{\boldsymbol{\theta}}^{\boldsymbol{y}_i|\tilde{\boldsymbol{z}}_i}(\tilde{\boldsymbol{z}}_i)^{-\text{H}}$. This can be done for the CSGMM and the CSVAE equivalently. Also note that $\boldsymbol{C}_{\boldsymbol{\theta}}^{\boldsymbol{s}|\boldsymbol{y}_i,\tilde{\boldsymbol{z}}_i}(\tilde{\boldsymbol{z}}_i)$ in (25) does not have to computed explicitly for training the CSVAE (cf. Appendix A.4). Moreover, for training the CSGMM, we solely require its diagonal entries (cf. (34)).

## A.4. Detailed Description of Closed Forms for the CSVAE Objective

The objective (12) for CSVAEs consists of three terms, i.e., $\mathbb{E}_{p_{\boldsymbol{\theta}}(\boldsymbol{s}|\tilde{\boldsymbol{z}}_i,\boldsymbol{y}_i)}[\log p(\boldsymbol{y}_i|\boldsymbol{s})]$, $\text{D}_{\text{KL}}(q_{\boldsymbol{\phi}}(\boldsymbol{z}|\boldsymbol{y}_i)||p(\boldsymbol{z}))$ and $\text{D}_{\text{KL}}(p_{\boldsymbol{\theta}}(\boldsymbol{s}|\tilde{\boldsymbol{z}}_i,\boldsymbol{y}_i)||p_{\boldsymbol{\theta}}(\boldsymbol{s}|\tilde{\boldsymbol{z}}_i))$. The second is given by (Kingma & Welling, 2014)

$$\text{D}_{\text{KL}}(q_{\boldsymbol{\phi}}(\boldsymbol{z}|\boldsymbol{y}_i)||p(\boldsymbol{z})) = -\frac{1}{2}\sum_{j=1}^{N_L}(1 + \log\sigma_{j,\boldsymbol{\phi}}^2(\boldsymbol{y}_i) - \mu_{j,\boldsymbol{\phi}}(\boldsymbol{y}_i) - \sigma_{j,\boldsymbol{\phi}}^2(\boldsymbol{y}_i))\tag{27}$$

with $q_{\boldsymbol{\phi}}(\boldsymbol{z}|\boldsymbol{y}_i) = \mathcal{N}(\boldsymbol{z};\boldsymbol{\mu}_{\boldsymbol{\phi}}(\boldsymbol{y}_i),\text{diag}(\boldsymbol{\sigma}_{\boldsymbol{\phi}}^2(\boldsymbol{y}_i)))$ and $\mu_{j,\boldsymbol{\phi}}(\boldsymbol{y}_i)$ and $\sigma_{j,\boldsymbol{\phi}}^2(\boldsymbol{y}_i))$ being the $j$th entry of $\boldsymbol{\mu}_{\boldsymbol{\phi}}(\boldsymbol{y}_i)$ and $\boldsymbol{\sigma}_{\boldsymbol{\phi}}^2(\boldsymbol{y}_i)$, respectively. Moreover, $N_L$ is the CSVAE's latent dimension. Separate closed-form solutions for the first and third term in the real-valued case with constant noise variance are given in (Böck et al., 2024b). When adjusting the derivation to the complex-valued case with varying noise covariances $\sigma_i^2$, both terms equal

$$\mathbb{E}_{p_{\boldsymbol{\theta}}(\boldsymbol{s}|\tilde{\boldsymbol{z}}_i,\boldsymbol{y}_i)}[\log p(\boldsymbol{y}_i|\boldsymbol{s})] = -\left(M\log(\pi\sigma^2) + \frac{1}{\sigma_i^2}\left(\|\boldsymbol{y}_i - \boldsymbol{A}\boldsymbol{D}\boldsymbol{\mu}_{\boldsymbol{\theta}}^{\boldsymbol{s}|\boldsymbol{y}_i,\tilde{\boldsymbol{z}}_i}(\tilde{\boldsymbol{z}}_i)\|_2^2 + \text{tr}(\boldsymbol{A}\boldsymbol{D}\boldsymbol{C}_{\boldsymbol{\theta}}^{\boldsymbol{s}|\boldsymbol{y}_i,\tilde{\boldsymbol{z}}_i}(\tilde{\boldsymbol{z}}_i)\boldsymbol{D}^{\text{H}}\boldsymbol{A}^{\text{H}})\right)\right)\tag{28}$$

and

$$\begin{aligned}\text{D}_{\text{KL}}(p_{\boldsymbol{\theta}}(\boldsymbol{s}|\tilde{\boldsymbol{z}}_i,\boldsymbol{y}_i)||p_{\boldsymbol{\theta}}(\boldsymbol{s}|\tilde{\boldsymbol{z}}_i)) = &\left(\log\det\left(\text{diag}(\boldsymbol{\gamma}_{\boldsymbol{\theta}}(\tilde{\boldsymbol{z}}_i))\right) - \log\det\left(\boldsymbol{C}_{\boldsymbol{\theta}}^{\boldsymbol{s}|\boldsymbol{y}_i,\tilde{\boldsymbol{z}}_i}(\tilde{\boldsymbol{z}}_i)\right) - S\right.\\ &\left. + \text{tr}\left(\text{diag}\left((\boldsymbol{\gamma}_{\boldsymbol{\theta}}(\tilde{\boldsymbol{z}}_i)^{-1})\,\boldsymbol{C}_{\boldsymbol{\theta}}^{\boldsymbol{s}|\boldsymbol{y}_i,\tilde{\boldsymbol{z}}_i}(\tilde{\boldsymbol{z}}_i)\right) + \boldsymbol{\mu}_{\boldsymbol{\theta}}^{\boldsymbol{s}|\boldsymbol{y}_i,\tilde{\boldsymbol{z}}_i}(\tilde{\boldsymbol{z}}_i)^{\text{H}}\text{diag}\left(\boldsymbol{\gamma}_{\boldsymbol{\theta}}(\tilde{\boldsymbol{z}}_i)^{-1}\right)\boldsymbol{\mu}_{\boldsymbol{\theta}}^{\boldsymbol{s}|\boldsymbol{y}_i,\tilde{\boldsymbol{z}}_i}(\tilde{\boldsymbol{z}}_i)\right)\end{aligned}\tag{29}$$

where $\boldsymbol{\mu}_{\boldsymbol{\theta}}^{\boldsymbol{s}|\boldsymbol{y}_i,\tilde{\boldsymbol{z}}_i}(\tilde{\boldsymbol{z}}_i)$ and $\boldsymbol{C}_{\boldsymbol{\theta}}^{\boldsymbol{s}|\boldsymbol{y}_i,\tilde{\boldsymbol{z}}_i}(\tilde{\boldsymbol{z}}_i)$ are the mean and covariance matrix of $p_{\boldsymbol{\theta}}(\boldsymbol{s}|\tilde{\boldsymbol{z}}_i,\boldsymbol{y}_i)$ (cf. (24) and (25)). Moreover, (Böck et al., 2024b) reformulates both terms and shows that they partially cancel out, leading to a more efficient computation, i.e.,

$$\begin{aligned}\mathbb{E}_{p_{\boldsymbol{\theta}}(\boldsymbol{s}|\tilde{\boldsymbol{z}}_i,\boldsymbol{y}_i)}[\log p(\boldsymbol{y}_i|\boldsymbol{s})] - \text{D}_{\text{KL}}(p_{\boldsymbol{\theta}}(\boldsymbol{s}|\tilde{\boldsymbol{z}}_i,\boldsymbol{y}_i)||p_{\boldsymbol{\theta}}(\boldsymbol{s}|\tilde{\boldsymbol{z}}_i)) = &-\left(M\log(\pi\sigma_i^2) + \frac{1}{\sigma_i^2}\left(\|\boldsymbol{y}_i - \boldsymbol{A}\boldsymbol{D}\boldsymbol{\mu}_{\boldsymbol{\theta}}^{\boldsymbol{s}|\boldsymbol{y}_i,\tilde{\boldsymbol{z}}_i}(\tilde{\boldsymbol{z}}_i)\|_2^2\right)\right)\\ &-\left(-M\log\sigma_i^2 + \log\det\boldsymbol{C}_{\boldsymbol{\theta}}^{\boldsymbol{y}|\tilde{\boldsymbol{z}}_i}(\tilde{\boldsymbol{z}}_i) + \boldsymbol{\mu}_{\boldsymbol{\theta}}^{\boldsymbol{s}|\boldsymbol{y}_i,\tilde{\boldsymbol{z}}_i}(\tilde{\boldsymbol{z}}_i)^{\text{H}}\text{diag}\left(\boldsymbol{\gamma}_{\boldsymbol{\theta}}(\tilde{\boldsymbol{z}}_i)^{-1}\right)\boldsymbol{\mu}_{\boldsymbol{\theta}}^{\boldsymbol{s}|\boldsymbol{y}_i,\tilde{\boldsymbol{z}}_i}(\tilde{\boldsymbol{z}}_i)\right)\end{aligned}\tag{30}$$

Note that due to having complex-valued Gaussians, (30) differs slightly from the terms in (Böck et al., 2024b).

## A.5. Detailed Explanation of the E-step and M-step for the CSGMM

**E-step**  Generally, the E-step corresponds to computing the model's posterior $p(\boldsymbol{s},k|\boldsymbol{y}_i) = p(\boldsymbol{s}|k,\boldsymbol{y}_i)p(k|\boldsymbol{y}_i)$ for each training sample $\boldsymbol{y}_i$. Equivalent to the CSVAE, $p(\boldsymbol{s}|k,\boldsymbol{y}_i)$ is Gaussian whose mean and covariance can be computed via (24) and (25). Moreover, $p(k|\boldsymbol{y}_i)$ can be calculated using Bayes, i.e.,

$$p(k|\boldsymbol{y}_i) = \frac{p(\boldsymbol{y}_i|k)p(k)}{\sum_k p(\boldsymbol{y}_i|k)p(k)}.\tag{31}$$

Due to modeling $\boldsymbol{s}|k$ to be a zero-mean Gaussian and $\boldsymbol{y}$ being linearly dependent on $\boldsymbol{s}$ (cf. Section 3.3), the distribution $p(\boldsymbol{y}_i|k)$ is a zero-mean Gaussian with covariance matrix (cf. (26))

$$\boldsymbol{C}_k^{\boldsymbol{y}_i|k} = \boldsymbol{A}\boldsymbol{D}\text{diag}(\boldsymbol{\gamma}_k)\boldsymbol{D}^{\text{H}}\boldsymbol{A}^{\text{H}} + \sigma_i^2\,\mathbf{I}.\tag{32}$$

Thus, (31) is computable in closed form.

**M-step**  The M-step for the $(u+1)$th iteration for CSGMMs solves

$$\{\rho_{k,(u+1)}, \gamma_{k,(u+1)}\} = \underset{\{\rho_k, \gamma_k\}}{\operatorname{argmax}} \sum_{\boldsymbol{y}_i \in \mathcal{Y}} \mathbb{E}_{p_u(\boldsymbol{s},k|\boldsymbol{y}_i)} \left[\log p(\boldsymbol{y}_i, \boldsymbol{s}, k)\right] \quad \text{s.t.} \sum_k \rho_k = 1. \tag{33}$$

with $p_u(\boldsymbol{s}, k|\boldsymbol{y}_i)$ being the posterior for the $i$th training sample computed in the $u$th iteration of the EM algorithm. The closed-form solution is given by (Böck et al., 2024b)

$$\boldsymbol{\gamma}_{k,(u+1)} = \frac{\sum_{\boldsymbol{y}_i \in \mathcal{Y}} p_u(k|\boldsymbol{y}_i)(|\boldsymbol{\mu}_{k,u}^{\boldsymbol{s}|\boldsymbol{y}_i,k}|^2 + \operatorname{diag}(\boldsymbol{C}_{k,u}^{\boldsymbol{s}|\boldsymbol{y}_i,k}))}{\sum_{\boldsymbol{y}_i \in \mathcal{Y}} p_u(k|\boldsymbol{y}_i)} \tag{34}$$

$$\rho_{k,(u+1)} = \frac{\sum_{\boldsymbol{y}_i \in \mathcal{Y}} p_u(k|\boldsymbol{y}_i)}{|\mathcal{Y}|}. \tag{35}$$

Note that for improving training stability, one can clip $\boldsymbol{\gamma}_{k,(u+1)}$ from below at, e.g., $10^{-7}$.

### A.6. Detailed Explanation for the Guarantee of Physical Consistency by SBGMs

We require three lines of argumentation to fully explain why the statistical model in (6)-(8) aligns with the inherent properties of wireless channels and, thus, is suited for generating wireless channels.

- The complex path losses $\rho_{\ell,m}$ in (1) model several physical as well as technical effects contributing to the wireless channel, such as the antennas' radiation patterns or changes in polarization due to reflections (Jaeckel et al., 2023). Additionally, their phases also contain the phase shift due to the center frequency modulation, i.e., $2\pi f_c \tau_{\ell,m}$ with $f_c$ being the center frequency (Tse & Viswanath, 2005). We reformulate $f_c \tau_{\ell,m} = d_{\ell,m}/\lambda_c$ with $d_{\ell,m}$ being the (sub-)path length between the transmitter and receiver (cf. (1)), and $\lambda_c$ being the corresponding wavelength (i.e., $c = \lambda_c f_c$ with the speed of light $c$). Typical center frequencies $f_c$ for the wireless transmission of signals range from a few GHz to tens or even hundreds of GHz. Thus, the corresponding wavelength $\lambda_c$ takes values of at most a few centimeters or smaller. Consequently, by just changing the (sub-)path length by $\lambda_c$ (i.e., a few centimeters or less), the path loss phase $\arg(\rho_{\ell,m})$ takes a full turn of $2\pi$. In other words, by just slight movements of users, the path loss phases $\arg(\rho_{\ell,m})$ rapidly change. Since we want to statistically describe the wireless channel over a whole scenario whose dimensions are much larger than the center wavelength and, thus, lead to $d_{\ell,m} \gg \lambda_c$, the path loss phases $\arg(\rho_{\ell,m})$ are generally modeled to be uniformly distributed (Tse & Viswanath, 2005).

- When choosing the dimension of the latent variable $\boldsymbol{z}$ in CGLMs (e.g., VAEs) to be smaller than the dimension of the data whose distribution the CGLM is supposed to be learned, the training of the CGLM aims $\boldsymbol{z}$ to be a statistically meaningful (lower-dimensional) representation of the data of interest (Bishop & Bishop, 2023).

- By building on a probabilistic graph representation of wireless channels, the work in (Böck et al., 2024a) proves that if a variable does not contain any information about the complex path loss phases $\arg(\rho_{\ell,m})$, the conditioning on this variable preserves the structural properties of channel moments, i.e., the channel's zero mean and Toeplitz covariance structure.

By combining these three arguments, (Böck et al., 2024a) reasons and experimentally validates that the latent variable $\boldsymbol{z}$ of CGLMs is trained to not capture $\arg(\rho_{\ell,m})$-related information as these phases do not contain distinct statistically characteristic channel features. In other words, (Böck et al., 2024a) shows that the statistical model of CGLMs, i.e.,

$$p_{\boldsymbol{\theta}, \boldsymbol{\delta}}(\boldsymbol{x}) = \int p_{\boldsymbol{\theta}}(\boldsymbol{x}|\boldsymbol{z}) p_{\boldsymbol{\delta}}(\boldsymbol{z}) \mathrm{d}\boldsymbol{z} = \int \mathcal{N}_{\mathbb{C}}(\boldsymbol{x}; \boldsymbol{\mu}_{\boldsymbol{\theta}}(\boldsymbol{z}), \boldsymbol{C}_{\boldsymbol{\theta}}(\boldsymbol{z})) p_{\boldsymbol{\delta}}(\boldsymbol{z}) \mathrm{d}\boldsymbol{z} \tag{36}$$

is learned to fulfill $\boldsymbol{\mu}_{\boldsymbol{\theta}}(\boldsymbol{z}) = \boldsymbol{0}$ and $\boldsymbol{C}_{\boldsymbol{\theta}}(\boldsymbol{z})$ being (block-)Toeplitz structured when being properly trained on channel realizations.

In SBGM, we use (7) and (8) as the statistical model for the compressible representation $\boldsymbol{s}_{\mathrm{R}}$ (or $\boldsymbol{s}_{t,f}$). By that, we implicitly model $\boldsymbol{h}$ to be distributed according to a CGLM with

$$p_{\boldsymbol{\theta}, \boldsymbol{\delta}}(\boldsymbol{h}) = \int p_{\boldsymbol{\theta}}(\boldsymbol{h}|\boldsymbol{z}) p_{\boldsymbol{\delta}}(\boldsymbol{z}) \mathrm{d}\boldsymbol{z} = \int \mathcal{N}_{\mathbb{C}}(\boldsymbol{h}; \boldsymbol{0}, \boldsymbol{D}_{\mathrm{R}} \operatorname{diag}(\boldsymbol{\gamma}_{\boldsymbol{\theta}}(\boldsymbol{z})) \boldsymbol{D}_{\mathrm{R}}^{\mathrm{H}}) p_{\boldsymbol{\delta}}(\boldsymbol{z}) \mathrm{d}\boldsymbol{z} \tag{37}$$

and $\boldsymbol{D}_{\mathrm{R}}\mathrm{diag}(\boldsymbol{\gamma}_{\boldsymbol{\theta}}(\boldsymbol{z}))\boldsymbol{D}_{\mathrm{R}}$ being Toeplitz (or block-Toeplitz when using $\boldsymbol{D}_{t,f}$), which perfectly aligns with the findings from (Böck et al., 2024a).

## B. Limitations

Our proposed methods exhibit some limitations, which are discussed below.

**Non-Stationary Channel Generation** When modeling the wireless channel using its compressible representation (5) and (20), one assumption is that each dictionary column only depends on one single grid point. More precisely, in SIMO, each column $\boldsymbol{a}_{\mathrm{R}}(\frac{g\pi}{S_{\mathrm{R}}})$ corresponds to only one single angle $\frac{g\pi}{S_{\mathrm{R}}}$. From a physical perspective, this means that at each antenna, the impinging waveform comes from the exact same direction. This is called the far-field approximation and is a common assumption in radar and wireless communication exploiting multiple antennas (Yin & Cheng, 2016). From a statistical perspective, the far-field approximation leads to a spatially stationary process. Equivalently, in OFDM, each dictionary column in (20) only corresponds to a single delay-doppler tuple $(j\overline{\tau}/S_f, i\overline{\vartheta}/S_t)$. Thus, each timestamp and subcarrier in the OFDM grid is modeled to experience the same delays and doppler shifts. This results in a stationary process in the time and frequency domain, referred to as the wide-sense-stationary-uncorrelated-scattering (WSSUS) assumption (Bello, 1963). While the stationary assumptions are common in radar and wireless communication, some applications require non-stationary channel characteristics, such as the generation of long channel trajectories, that our proposed method, in its current form, cannot model.

**Off-Grid Mismatches** Another limitation is that our method requires a discretization of the physical parameter space (cf. Section 3.2). In consequence, when a channel in the training dataset exhibits physical parameters that do not exactly match any grid point, our proposed method cannot distinguish between having a single path with parameters between two grid points and having two paths on neighboring grid points. This mismatch is controlled by the grid resolution.

**OFDM Pilot Pattern** A further limitation is specific for OFDM. It is well known that compressive sensing (CS) methods experience a decrease in performance when using selection matrices that correspond to a regular pilot pattern compared to using (pseudo-)random selection matrices (Gaudio et al., 2022). As the training in SBGM as well as the physics-informed *implicit* AmbientGAN builds on CS, this limitation also holds for our proposed methods and constrains the set of suitable OFDM pilot patterns.

## C. Model Extensions

### C.1. Customizing the Number of Paths per Channel Realizations

When modeling wireless channels using ray tracing, one possibility is to customize the number of paths per channel realization (Alkhateeb, 2019). This can be beneficial to, e.g., reduce computational complexity or only capture the most relevant effects in the channel realization of interest. Equivalent to ray tracing, our proposed method also allows the adjustment of the number of paths. More precisely, our method generates new complex-valued vectors $\boldsymbol{s}_{t,f}$ (or $\boldsymbol{s}_{\mathrm{R}}$) where each entry represents the complex-valued path loss corresponding to one gridpoint, i.e., one delay-doppler tuple or one angle. Thus, we can interpret each non-zero entry in $\boldsymbol{s}_{t,f}$ as a single path, where the corresponding index in $\boldsymbol{s}_{t,f}$ together with the complex-valued entry determines the path's physical parameters. Thus, when we want to restrict our generated channel realizations to only possess $p_{\max}$ paths, we can set all entries in the newly generated $\boldsymbol{s}_{t,f}$ (or $\boldsymbol{s}_{\mathrm{R}}$) to zero that do not belong to the $p_{\max}$ strongest entries in a squared absolute value sense.

### C.2. Reducing the Number of Learnable Parameters when Considering Multiple Domains

In case of considering multiple domains (e.g., the time and frequency domain in OFDM, cf. Section 3.4), one possibility to reduce the number of learnable parameters is to constrain the conditional covariance matrix $\mathbb{E}[\boldsymbol{h}\boldsymbol{h}^{\mathrm{H}}|\boldsymbol{z}]$ to not only be block-Toeplitz (cf. Section 4.3) but rather a Kronecker of Toeplitz matrices forming a subset of all block-Toeplitz matrices. In particular, by constraining $\boldsymbol{\gamma}_{\boldsymbol{\theta}}(\boldsymbol{z}) = \boldsymbol{\gamma}_{\boldsymbol{\theta}}^{(t)}(\boldsymbol{z}) \otimes \boldsymbol{\gamma}_{\boldsymbol{\theta}}^{(f)}(\boldsymbol{z})$ the resulting conditional channel covariance matrix $\boldsymbol{D}_{t,f}\mathrm{diag}(\boldsymbol{\gamma}_{\boldsymbol{\theta}}(\boldsymbol{z}))\boldsymbol{D}_{t,f}^{\mathrm{H}}$ in OFDM is given by

$$(\boldsymbol{D}_t \otimes \boldsymbol{D}_f)\,\mathrm{diag}(\boldsymbol{\gamma}_{\boldsymbol{\theta}}^{(t)}(\boldsymbol{z}) \otimes \boldsymbol{\gamma}_{\boldsymbol{\theta}}^{(f)}(\boldsymbol{z}))\,(\boldsymbol{D}_t \otimes \boldsymbol{D}_f)^{\mathrm{H}} \tag{38}$$

and, thus, can be written as a Kronecker product of Toeplitz covariance matrices. This approximation reduces the number of variance parameters learned from $S_t S_f$ to $S_t + S_f$. For the CSVAE, this constraint can be enforced by letting the NN output an $S_f$- and an $S_t$-dimensional vector and subsequently building the Kronecker product instead of outputting a $S_f S_t$-dimensional vector. For CSGMM, incorporating this constraint requires a new M-step. Specifically, the M-step with the constraint $\boldsymbol{\gamma}_k = \boldsymbol{\gamma}_k^{(t)} \otimes \boldsymbol{\gamma}_k^{(f)}$ exhibits no closed form and must be solved iteratively. In the following, we derive closed forms for the global solution of the update steps when applying coordinate search.

We aim solve (33) with the constraint $\boldsymbol{\gamma}_k = \boldsymbol{\gamma}_k^{(t)} \otimes \boldsymbol{\gamma}_k^{(f)}$. Following the derivation in Appendix F of (Böck et al., 2024b) and considering that we have complex-valued Gaussians, we reformulate the optimization problem as

$$\operatorname*{argmax}_{\{\rho_k, \boldsymbol{\gamma}_k\}} \sum_{\boldsymbol{y}_i \in \mathcal{Y}} \sum_{k=1}^{K} p_u(k|\boldsymbol{y}_i) \left( -\left( S \log \pi + \sum_{j=1}^{S} \left( \log \gamma_{k,j} + \frac{|\mu_{k,j}^{\boldsymbol{s}|\boldsymbol{y}_i,k}|^2 + C_{k,j,j}^{\boldsymbol{s}|\boldsymbol{y}_i,k}}{\gamma_{k,j}} \right) \right) + \log \rho_k \right)$$

$$\text{s.t.} \sum_{k} \rho_k = 1, \quad \boldsymbol{\gamma}_k = \boldsymbol{\gamma}_k^{(t)} \otimes \boldsymbol{\gamma}_k^{(f)} \text{ for all } k \tag{39}$$

with $S = S_t \cdot S_f$ and $\mu_{k,j}^{\boldsymbol{s}|\boldsymbol{y}_i,k}$ and $C_{k,j,j}^{\boldsymbol{s}|\boldsymbol{y}_i,k}$ denote the $j$th entry of $\boldsymbol{\mu}_{k,u}^{\boldsymbol{s}|\boldsymbol{y}_i,k}$ and the diagonal of $\boldsymbol{C}_{k,u}^{\boldsymbol{s}|\boldsymbol{y}_i,k}$ in the $u$th iteration, respectively. By defining $r(q,p) = (q-1)S_f + p$, we can rewrite

$$\gamma_{k,r(q,p)} = \gamma_{k,q}^{(t)} \cdot \gamma_{k,p}^{(f)} \tag{40}$$

and, thus, (39) can be rewritten as

$$\operatorname*{argmax}_{\{\rho_k, \boldsymbol{\gamma}_k\}} \sum_{\boldsymbol{y}_i \in \mathcal{Y}} \sum_{k=1}^{K} p_u(k|\boldsymbol{y}_i) \left( -S \log \pi - S_f \sum_{q=1}^{S_t} \log \gamma_{k,q}^{(t)} - S_t \sum_{p=1}^{S_f} \log \gamma_{k,p}^{(f)} - \sum_{q,p=1}^{S_t,S_f} \frac{|\mu_{k,r(q,p)}^{\boldsymbol{s}|\boldsymbol{y}_i,k}|^2 + C_{k,r(q,p),r(q,p)}^{\boldsymbol{s}|\boldsymbol{y}_i,k}}{\gamma_{k,q}^{(t)} \cdot \gamma_{k,p}^{(f)}} + \log \rho_k \right)$$

$$\text{s.t.} \sum_{k} \rho_k = 1 \tag{41}$$

The corresponding Lagrangian $\mathcal{L}$ is given by

$$\mathcal{L} = \sum_{\boldsymbol{y}_i \in \mathcal{Y}} \sum_{k=1}^{K} p_u(k|\boldsymbol{y}_i) \Bigg( -S \log \pi - S_f \sum_{q=1}^{S_t} \log \gamma_{k,q}^{(t)} - S_t \sum_{p=1}^{S_f} \log \gamma_{k,p}^{(f)} - \sum_{q,p=1}^{S_t,S_f} \frac{|\mu_{k,r(q,p)}^{\boldsymbol{s}|\boldsymbol{y}_i,k}|^2 + C_{k,r(q,p),r(q,p)}^{\boldsymbol{s}|\boldsymbol{y}_i,k}}{\gamma_{k,q}^{(t)} \cdot \gamma_{k,p}^{(f)}} +$$

$$\log \rho_k \Bigg) + \nu \Big( 1 - \sum_{k} \rho_k \Big) \tag{42}$$

with Lagrangian multiplier $\nu$. In the following, we consider coordinate search, i.e., we keep $\{\boldsymbol{\gamma}_k^{(f)}\}_{k=1}^{K}$ fixed and solely optimize over $\{\boldsymbol{\gamma}_k^{(t)}\}_{k=1}^{K}$ (and vice versa). By taking the derivative of $\mathcal{L}$ with respect to $\gamma_{\overline{k},\overline{q}}^{(t)}$ and setting it to zero, we end up with

$$\frac{\partial}{\partial \gamma_{\overline{k},\overline{q}}^{(t)}} \mathcal{L} = \sum_{\boldsymbol{y}_i \in \mathcal{Y}} p_u(\overline{k}|\boldsymbol{y}_i) \left( -\frac{S_f}{\gamma_{\overline{k},\overline{q}}^{(t)}} + \sum_{p=1}^{S_f} \frac{|\mu_{\overline{k},r(\overline{q},p)}^{\boldsymbol{s}|\boldsymbol{y}_i,k}|^2 + C_{\overline{k},r(\overline{q},p),r(\overline{q},p)}^{\boldsymbol{s}|\boldsymbol{y}_i,k}}{\gamma_{\overline{k},\overline{q}}^{(t)2} \gamma_{\overline{k},p}^{(f)}} \right) = 0 \tag{43}$$

and, thus,

$$\gamma_{\overline{k},\overline{q}}^{(t)} = \sum_{p=1}^{S_f} \frac{\sum_{i=1}^{N_t} p_u(\overline{k}|\boldsymbol{y}_i) \left( |\mu_{r(\overline{q},p)}^{\boldsymbol{s}|\boldsymbol{y}_i,\overline{k}}|^2 + C_{r(\overline{q},p),r(\overline{q},p)}^{\boldsymbol{s}|\boldsymbol{y}_i,\overline{k}} \right)}{S_f \gamma_{\overline{k},p}^{(f)} \sum_{i=1}^{N_t} p(\overline{k}|\boldsymbol{y}_i)} \tag{44}$$

for all $\overline{q} = 1, \ldots, S_t$, $\overline{k} = 1, \ldots, K$. Due to the symmetry of (41) with respect to $\boldsymbol{\gamma}_k^{(t)}$ and $\boldsymbol{\gamma}_k^{(f)}$, we find an equivalent update for $\boldsymbol{\gamma}_k^{(f)}$, i.e.,

$$\gamma_{\overline{k},\overline{p}}^{(f)} = \sum_{q=1}^{S_t} \frac{\sum_{i=1}^{N_t} p_u(\overline{k}|\boldsymbol{y}_i) \left( |\mu_{r(q,\overline{p})}^{\boldsymbol{s}|\boldsymbol{y}_i,\overline{k}}|^2 + C_{r(q,\overline{p}),r(q,\overline{p})}^{\boldsymbol{s}|\boldsymbol{y}_i,\overline{k}} \right)}{S_t \gamma_{\overline{k},q}^{(t)} \sum_{i=1}^{N_t} p(\overline{k}|\boldsymbol{y}_i)} \tag{45}$$

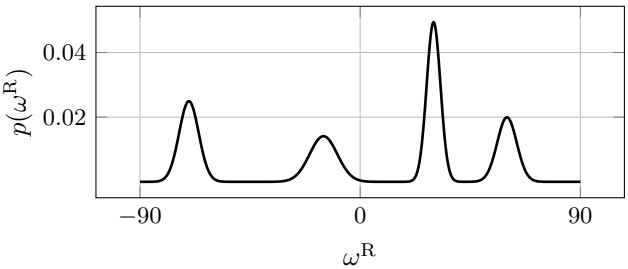

Figure 7. Distribution $p(\omega^{\mathrm{R}})$ of the path angle in the modified 3GPP dataset.

for all $\overline{p} = 1, \ldots, S_f, \overline{k} = 1, \ldots, K$. We suppressed the index for the coordinate search due to readability. The update steps of the weights $\{\rho_k\}_{k=1}^K$ equal those in (Böck et al., 2024b) and are given by

$$\rho_{\overline{k}} = \frac{\sum_{\boldsymbol{y}_i \in \mathcal{Y}} p_u(\overline{k}|\boldsymbol{y}_i)}{\nu} \tag{46}$$

$$\nu = |\mathcal{Y}| \tag{47}$$

## D. Dataset Description and Pre-Processing

### D.1. Modified 3GPP Dataset Description and Pre-Processing

Our modified 3GPP dataset is based on the conditionally normal channels described in (Neumann et al., 2018). More precisely, each channel realization is generated following two steps. First, we draw one random angle $\omega^{(\mathrm{R})}$ from the distribution illustrated in Fig. 7. Subsequently, we draw the channel $\boldsymbol{h}$ from $\mathcal{N}(\boldsymbol{h}; \boldsymbol{0}, \boldsymbol{C}_{\omega^{(\mathrm{R})}})$, where

$$\boldsymbol{C}_{\omega^{(\mathrm{R})}} = \int_{-\pi}^{\pi} g(\theta; \omega^{\mathrm{R}}) \boldsymbol{a}_{\mathrm{R}}(\theta) \boldsymbol{a}_{\mathrm{R}}(\theta)^{\mathrm{H}} \mathrm{d}\theta \tag{48}$$

and $g(\theta; \omega^{(\mathrm{R})})$ is a Laplacian with mean $\omega^{(\mathrm{R})}$ and standard deviation of 2 degree. Moreover, $\boldsymbol{a}_{\mathrm{R}}(\cdot)\big|_i = \mathrm{e}^{-\mathrm{j}\,\pi(i-1)\sin(\cdot)}$ corresponding to a ULA with equidistant antenna spacing (cf. Section 3.2). The angle distribution in Fig. 7 artificially represents a scenario with, e.g., four street canyons where the users' positions are mainly distributed in four different angular regions. The 2-degree standard deviation is in line with the 3GPP standard and referred to as per-path angle spread (3GPP, 2024b). It simulates a small spread of the main angle $\omega^{(\mathrm{R})}$ due to scatterers close to the users. We apply no other pre-processing. We define the SNR in dB to be $\mathrm{SNR} = 10\log_{10}(\mathbb{E}[\|\boldsymbol{h}\|^2]/(\sigma^2 N))$, where $N$ is the dimension of $\boldsymbol{h}$, i.e., the number of antennas, $\sigma^2$ is the noise variance, and $\mathbb{E}[\|\boldsymbol{h}\|^2]$ is estimated using 10000 samples generated in the discussed manner. For producing the training dataset, we draw $\mathrm{SNR}_i$ uniformly between 0dB and 20dB for each training sample and compute the corresponding noise variance $\sigma_i^2$. Subsequently, we generate the training dataset

$$\mathcal{Y} = \{\boldsymbol{y}_i \mid \boldsymbol{y}_i = \boldsymbol{h}_i + \boldsymbol{n}_i\} \tag{49}$$

with $\boldsymbol{n}_i \sim \mathcal{N}_{\mathbb{C}}(\boldsymbol{0}, \sigma_i^2\,\mathbf{I})$.

### D.2. QuaDRiGa Dataset Description and Pre-Processing

QuaDRiGa is a freely accessible geometry-based stochastic simulation platform for wireless channels (Jaeckel et al., 2023). The two QuaDRiGa datasets considered in our work build on the "3GPP_38.901_UMa" and the "3GPP_38.901_URa" scenario, which simulate an urban and a rural macrocell environment with parameters consistent with the 3GPP standard, respectively (Jaeckel et al., 2014). For the former, we use a setup with users in LOS as well as NLOS, as well as indoor and outdoor. The scenario contains three streets in a 120-degree sector of an outdoor cellular network. An illustration of the scenario is given in Fig. 8. The specific scenario parameters are given in Table 2. For the latter, we use a setup with users all in LOS and outdoor. An illustration of the scenario is given in Fig. 9. The specific scenario parameters are given in Table 3.

For both scenarios, we simulate OFDM channels with system configuration specified in Table 4. We denote these datasets with 5G-Urban and 5G-Rural, respectively. For evaluating the generalization performance of our proposed models, we

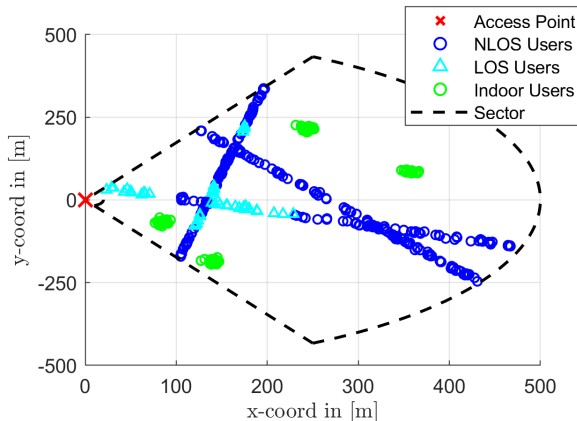

*Figure 8.* Urban QuaDRiGa scenario with three streets and regions with in- and outdoor as well as LOS and NLOS users.

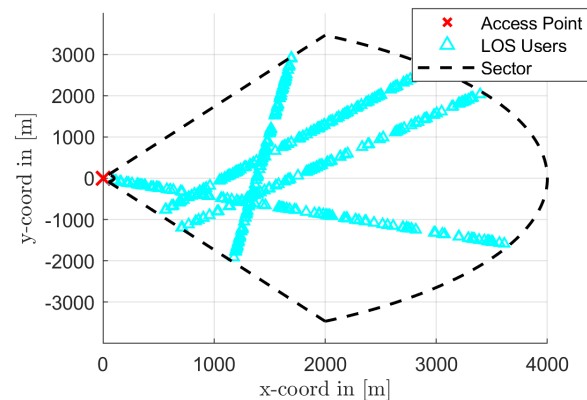

*Figure 9.* Rural QuaDRiGa scenario with four streets and only LOS users.

*Table 2.* Specific scenario parameters for the urban scenario.

| Distance Range User to BS | Velocity Range | Number Streets | In- to Outdoor User Ratio |
|---|---|---|---|
| 20-500m | 0-50km/h | 3 | 1/4 |

*Table 3.* Specific scenario parameters for the rural scenario.

| Distance Range User to BS | Velocity Range | Number Streets | In- to Outdoor User Ratio |
|---|---|---|---|
| 100-4000m | 70-100km/h | 4 | 0 |

*Table 4.* Specific parameters for the `5G` system configuration.

| Bandwidth | Subcarrier Spacing | Slot Duration | Symbol Duration |
|---|---|---|---|
| 360 kHz | 15 kHz | 1 ms | 1/14 ms |

*Table 5.* Specific parameters for the `Large` system config.

| Bandwidth | Subcarrier Spacing | Slot Duration | Symbol Duration |
|---|---|---|---|
| 1200 kHz | 60 kHz | 5.14 ms | 1/3.5 ms |

also simulate OFDM channels in the urban scenario with changed system configuration, specified in Table 5. The resulting dataset is denoted as `Large-Urban`.

In all cases, we pre-process the data before training the generative model. More specifically, we apply the common scaling of each channel realization by its effective path gain (PG), which models the attenuation due to the distance of the user and the BS as well as shadowing effects (Jaeckel et al., 2023). Subsequently, we scale the training dataset such that the estimated signal energy $\mathbb{E}[\|\boldsymbol{h}\|^2]$ equals the number of resource elements in the corresponding OFDM grid (i.e., 336 for the `5G` system configuration and 360 for the `Large` system configuration). For each tuple of number $M$ of pilots and system configuration (e.g., $(30, 5\mathrm{G})$), we draw one random selection matrix as measurement matrix $\boldsymbol{A} \in \mathbb{R}^{M \times N}$ (cf. (9)) and keep this matrix fixed for all training and validation samples. This selection matrix extracts $M$ random elements from the $N$-dimensional channel $\boldsymbol{h}$. Equivalent to the modified 3GPP dataset, we draw $\mathrm{SNR}_i$ uniformly between 5dB and 20dB for each training sample and compute the corresponding noise variance $\sigma_i^2 = \mathbb{E}[\|\boldsymbol{A}\boldsymbol{h}\|^2]/(M \cdot 10^{0.1\mathrm{SNR}_i[\mathrm{dB}]})$. Then, the datasets are given by

$$\mathcal{Y} = \{\boldsymbol{y}_i \mid \boldsymbol{y}_i = \boldsymbol{A}\boldsymbol{h}_i + \boldsymbol{n}_i\} \tag{50}$$

with $\boldsymbol{n}_i \sim \mathcal{N}_\mathbb{C}(\boldsymbol{0}, \sigma_i^2\,\boldsymbol{I})$.

### D.3. DeepMIMO Dataset Description and Pre-Processing

The modified 3GPP dataset (cf. Appendix D.1), as well as the QuaDRiGa dataset (cf. Appendix D.2), involve statistics in their channel generation process, due to, e.g., sampling underlying angles (modified 3GPP) or randomly placing scatterers in a simulated environment (QuaDRiGa). In addition to these channel models, we evaluate our method for SIMO based on a purely deterministic channel model, i.e., ray tracing. Next to being purely deterministic, another key difference between ray tracing and standardized channel models (and QuaDRiGa) is the absence of subpaths. More concretely, $M_\ell$ in (1) is set to 1 for all $\ell$ (Alkhateeb, 2019). Subpaths typically originate from minor effects, such as the spread of an impinging waveform at a rough surface into several paths with similar properties. Since these phenomena are difficult to simulate deterministically, they are left out in ray tracing. As described in Section 1, ray tracing requires a 3D replica of the environment of interest and the material properties within the scene. The DeepMIMO dataset proposed in (Alkhateeb, 2019) is a benchmark dataset for

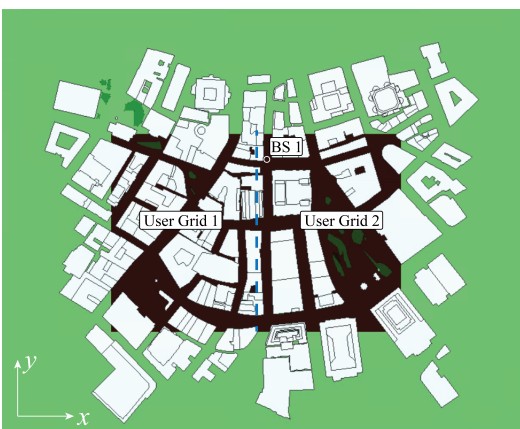

*Figure 10.* Ray tracing Boston scenario used in our simulation (cf. (Alkhateeb, 2019)).

wireless channel modeling, building on the ray tracing tool Remcom (Remcom) and offering several predefined scenarios. In our work, we use the Boston 5G scenario with 3.5GHz center frequency. In Fig. 10, we plot a 2D perspective of this scenario (Alkhateeb, 2019). We only consider users in "User Grid 2" since most of the users in "User Grid 1" are blocked and have no connection to the BS. DeepMIMO offers the possibility to simulate channels at gridpoints over the whole scenario in Fig. 10, where the spacing between two adjacent users is 37cm. In a first step towards the used dataset in our work, we exclude all users in "User Grid 2", which are blocked and have no connection to the BS, i.e., we only consider non-zero channels (either LOS or NLOS) in our dataset. Furthermore, we consider single antenna users and the BS is equipped with a ULA with $N$ antennas and $\lambda/2$ antenna spacing. This ULA is first placed along the x-axis and then rotated by 45 degrees around the z-axis. Moreover, we allow up to 8 paths to be simulated per channel realization.

We pre-process the data before training the generative models. Since the simulated channels in ray tracing involve large-scale fading (e.g., path losses due to the distance between the user and BS), their norms range over several orders of magnitude, making them ill-suited to directly train ML models. DeepMIMO provides the absolute values of each path's path loss (denoted as `DeepMIMO_dataset{i}.user{j}.path_params.power` in DeepMIMO Version v2, and denoted as $p_\ell^{\mathrm{DM}}$ in the following). Similar to how QuaDRiGa distinguishes between large-scale- and small-scale-fading (cf. (Jaeckel et al., 2023)), we scale each channel individually by $\sqrt{1/(\sum_\ell p_\ell^{\mathrm{DM}})}$.[8] Subsequently, we scale the training dataset such that the estimated signal energy $\mathbb{E}[\|\boldsymbol{h}\|^2]$ equals the number of antennas, i.e., $N$. We then apply the same method to compute noise variances $\sigma_i^2$ for each training channel realization as in Appendix D.1 to construct the used dataset. We consider the case $N = 32$ (and $N = 256$ for computing ground truth, cf. Appendix E), and denote the dataset as $\mathcal{Y}_{\mathrm{DeepM}}^{(32)}$.

## E. Architectures, Baselines, and Hyperparameters

**CSVAE**    For the CSVAE encoder, we use a simple fully-connected NN with ReLU activation function (cf. Table 6). We tested different architectures for the decoder, from which a deep decoder-motivated architecture performed the best (cf. (Heckel & Hand, 2019)). Specifically, the decoder contains two fully connected layers with ReLU activation followed by several blocks consisting of a convolutional layer with a kernel size of 1, a ReLU activation, and a (bi-)linear upsampling operation. For the simulations on OFDM, we incorporated 3 of these blocks (cf. Table 7), while for the modified 3GPP and DeepMIMO dataset (cf. Table 8), we used 2 of them. The final layer is a convolutional layer with a kernel size of 1. We applied hyperparameter tuning for each simulation setup to adjust the width and the depth d for the encoder as well as the linear layer width, the number of convolutional channels in the decoder, and the learning rate. For that, we took the model resulting in the largest evidence lower bound (ELBO) over a validation set of 5000 samples. For the optimization, we used the Adam optimizer (Kingma & Ba, 2015).

---

[8]Note that this normalization of each individual channel requires an estimation of the attenuation of the channel based on the distance between the user and the BS, which typically changes very slowly. The alternative normalization for all channels to have norm 1, respectively, requires genie-knowledge of each channel realization for offline training as well as online operation, which is why we do not normalize in this manner.

*Table 6.* CSVAE.

| Encoder |
| --- |
| d×Linear |
| ReLU |
| 1×Linear |

*Table 7.* CSVAE (OFDM).

| Decoder |
| --- |
| 2×Linear |
| ReLU |
| 1×Unflatten |
| 3×Conv2D (kernel size=1) |
| ReLU |
| Upsample (scale= 2, bilinear) |
| 1×Conv2D (kernel size=1) |

*Table 8.* CSVAE (3GPP & DeepMIMO).

| Decoder |
| --- |
| 2×Linear |
| ReLU |
| 1×Unflatten |
| 2×Conv1D (kernel size=1) |
| ReLU |
| Upsample (scale=2, linear) |
| 1×Conv1D (kernel size=1) |

*Table 9.* Autoencoder.

| Encoder |
| --- |
| d×Conv2D |
| ReLU |
| 1×Linear |
| 1×Tanh |

*Table 10.* Autoencoder.

| Decoder |
| --- |
| 1×Linear |
| 1×Unflatten |
| d×ConvTranspose2D |
| ReLU |
| 1×Conv2D (kernel size=1) |

*Table 11.* GAN (OFDM).

| Generator |
| --- |
| 1×Linear |
| ReLU |
| 1×Unflatten |
| 2×Upsample (scale=2,nearest) |
| Conv2D |
| Batch Normalization |
| ReLU |
| 1×Conv2D |

*Table 12.* GAN (OFDM).

| Discriminator |
| --- |
| 3×Conv2D |
| LeakyReLU(0.2) |
| Dropout($p = 0.25$) |
| 1×Flatten |
| 1×Linear |

Note that we need to enforce the variances $\boldsymbol{\sigma}_\phi^2(\boldsymbol{y})$ at the output of the encoder as well as the variances $\boldsymbol{\gamma}_\theta(\boldsymbol{z})$ at the output of the decoder to be positive. To do so, we interpret the corresponding outputs of the final layer to represent the logarithm of these variances and apply an exponential function to the output before any further processing. For increased stability, we also recommend implementing a lower bound for the decoder output representing $\log \boldsymbol{\gamma}_\theta(\boldsymbol{z})$ (e.g., $-10$).

**CSGMM**  The only hyperparameter for CSGMM is the number $K$ of components. We tune this parameter by choosing $K$, leading to the largest log-likelihood over the training dataset.

**Autoencoder**  We use an autoencoder to apply the cross-validation method from (Baur et al., 2025; Xiao et al., 2022). The chosen architecture resembles the proposed architecture in (Rizzello & Utschick, 2021) and is given in Table 9 and 10. The kernel size, the stride, and the padding of the 2D convolutional layers in the encoder are set to 3, 2, and 1, respectively. The stride and the padding in the 2D transposed convolutional layers of the decoder are set to 2 and 1, and the kernel size is set to either 3 or 4, depending on the output dimension to be matched. The final 2D convolutional layer has a kernel size, stride, and padding of 1, 1, and 0. We used the same architecture for the simulations on DeepMIMO but with 1D convolutional layers instead of 2D ones. All autoencoders in all simulations exhibit a latent dimension of 16. We applied hyperparameter tuning to adjust the width of the linear layer, the number of convolutional blocks d, the channel size of the convolutional layers, as well as the learning rate based on a ground-truth validation dataset. For QuaDRiGa, we used 10000 validation channel realizations. For the optimization, we used the Adam optimizer (Kingma & Ba, 2015).

**AmbientGAN for $s$ and $h$**  For both GAN variants, we utilize the architecture from (Doshi et al., 2022), i.e., the generator consists of a single fully connected layer with ReLU activation, an unflatten operation and, subsequently, two blocks of a nearest-neighbor upsampling operation, a convolutional layer, a batch normalization and a ReLU activation (cf. Table 11). The final layer is a convolutional layer. The kernel size, the stride, and the padding of all convolutional layers are set to 3, 1, and 1, respectively. The discriminator contains three blocks of a convolutional layer, a LeakyReLU activation, and a dropout layer. Subsequently, it contains a flatten operation and a final linear layer. For the simulations on DeepMIMO, we use the same architecture but with 1D convolutional layers instead of 2D ones. The width of the linear layers and the channel number of the convolutional layers is determined by hyperparameter tuning. For the simulations on QuaDRiGa, we

Table 13. Generalization Performance for `5G-Urban` to `Large-Urban` and vice versa ($M = 30$, $N_t = 10000$).

| | CSGMM trained on `5G-Urban` | CSGMM trained on `Large-Urban` |
|---|---|---|
| nMSE for the `5G-Urban` config. | 0.00109 | 0.00096 |
| $\rho_c$ for the `5G-Urban` config. | 0.99756 | 0.99783 |
| nMSE for the `Large-Urban` config. | 0.00178 | 0.00155 |
| $\rho_c$ for the `Large-Urban` config. | 0.99675 | 0.99701 |

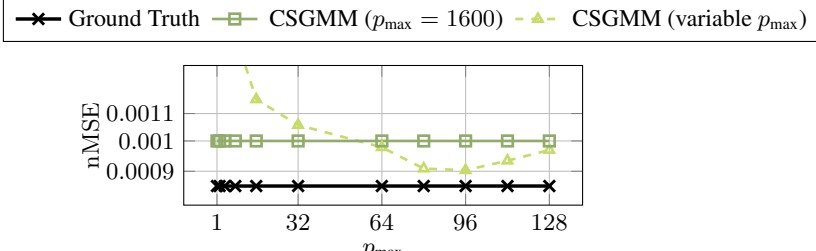

Figure 11. nMSE over the number of considered paths $p_{max}$ for the `5G-Urban` dataset with $M = 30$ and $N_t = 10\,000$.

take the autoencoder's reconstruction performance on a ground-truth validation set as the objective for the hyperparameter tuning. For parameter generation, we take the AmbientGAN leading to the smallest average generated angular spread and, thus, the best fitting histogram of spreads. In line with (Doshi et al., 2022), we used the RMSProp optimizer, adjust the learning rate via hyperparameter tuning, and the GAN variants are optimized using the Wasserstein GAN objective with gradient penalty explained in Appendix A.2.

**Ground-Truth Baseline for the Parameter Generation Performance** In Section 6.2, we evaluate the parameter generation performance for the 3GPP dataset and compare our method to ground truth. The same is done for the DeepMIMO dataset in Appendix 6.2. In general, for the simulation, we decide on a number of angle grid points $S$, set to 256. For computing the results for the ground-truth baseline, we artificially assume that we have as many antennas as gridpoints, i.e., 256. This results in $\boldsymbol{D}_R$ in (5) to be squared and invertible. In consequence, we can compute $\boldsymbol{s}_R$ for each channel realization in the test set and estimate the power angular profile (15) and the angular spread (16) using these calculated vectors.

## F. Additional Results

### F.1. Experiments for Validating the Generalization Performance

We analyze SBGM's capability to generalize to other system configurations without being retrained. More specifically, we train CSGMM using `5G-Urban` and $N_t = 10000$. After training, we use the dictionary $\boldsymbol{D}_{t,f}$ for the `Large-Urban` configuration, i.e., $\Delta f = 60\text{kHz}$, $\Delta T = \frac{1}{3.5}\text{ms}$ with $\boldsymbol{D}_{t,f} \in \mathbb{C}^{(20\cdot18)\times(40\cdot40)}$ (cf. Appendix A.1) to map generated $\boldsymbol{s}_{t,f}$ to new channel realizations that are different to the ones involved in the training dataset `5G-Urban`. Subsequently, we apply the same cross-validation method as explained in Section 6.2, i.e., we use these newly generated channel realizations to train an autoencoder whose performance is then evaluated on ground-truth QuaDRiGa channels in the `Large-Urban` configuration. We also do the same procedure vice versa, i.e., we train CSGMM using `Large-Urban` and $N_t = 10000$, and then adapt the dictionary to the `5G-Urban` configuration.

The results are given in Table 13. The underlined numbers indicate the performance where CSGMM has been trained on a different system configuration compared to the one used for channel generation. We can see that there is almost no difference between whether CSGMM has been trained on the system configuration that matches with the generated channel realizations afterward or a different one.

### F.2. Experiments for Controlling the Number of Paths Considered

In Fig. 11, we analyze the effect on the generated samples when varying $p_{max}$ (cf. Appendix C.1). More precisely, in Fig. 11, the nMSE is shown for varying $p_{max}$ where we used QuaDRiGa channels from `5G-Urban` (cf. Section D.2), $M = 30$ and $N_t = 10\,000$. Thus, we first train CSGMM with `5G-Urban`. We then generate 10000 new samples $\boldsymbol{s}_{t,f}$. We filter out all entries for each of these samples with a smaller squared absolute value than the $p_{max}$ strongest entries. We then map the

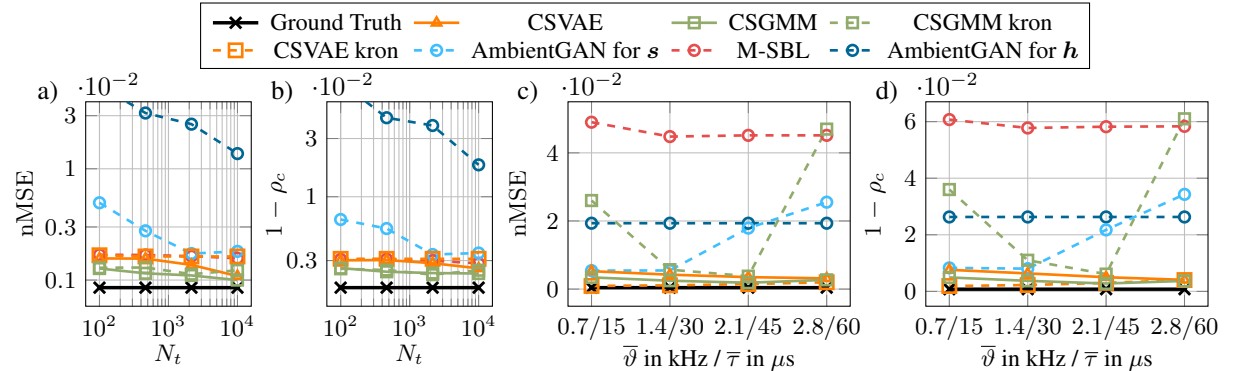

Figure 12. a) - d) nMSE and $\rho_c$ for reconstructing ground-truth channels by an autoencoder trained on channels $\boldsymbol{h} = \boldsymbol{D}\boldsymbol{s}$ produced by the trained models, respectively. In a) and b), we vary $N_t$ for the generative models with fixed $M = 30$ and dataset 5G-Urban, and in c) and d), we vary $\overline{\vartheta}$ and $\overline{\tau}$ with fixed $N_t = 3\,000$ and dataset 5G-Rural.

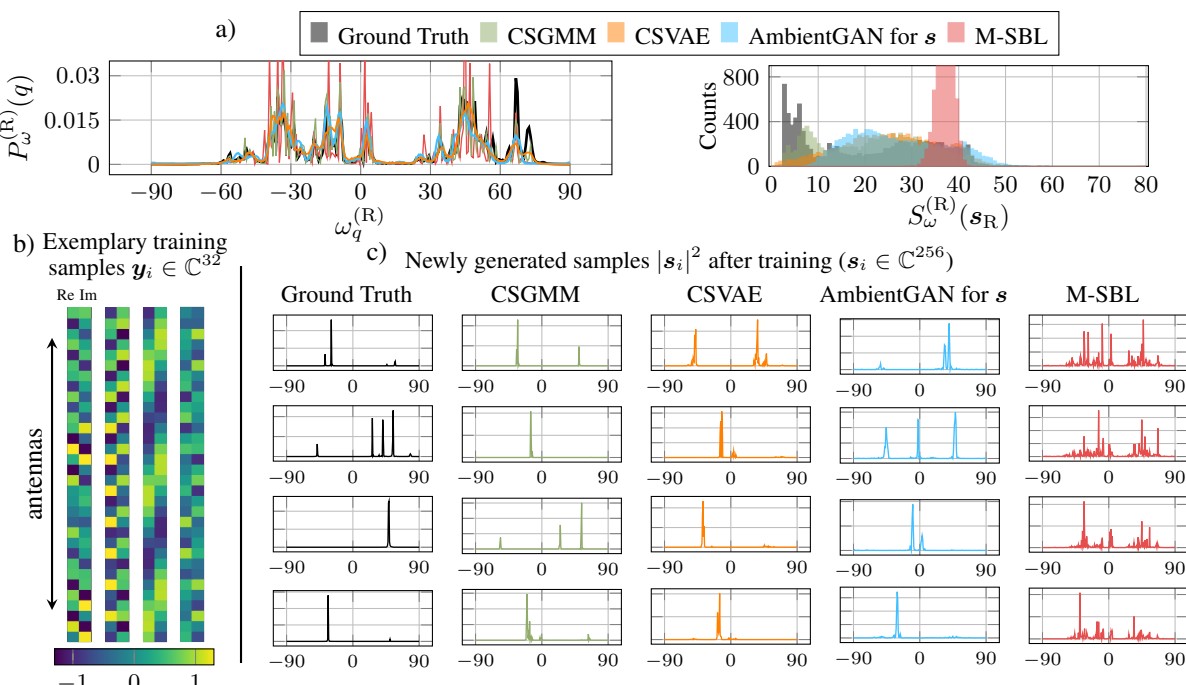

Figure 13. a) Power angular profile $P_\omega^{(\mathrm{R})}(q)$ and a histogram of the angular spread $S_\omega^{(\mathrm{R})}(\boldsymbol{s}_\mathrm{R})$ from 10000 generated samples by CSVAE, CSGMM, AmbientGAN for $\boldsymbol{s}$ and M-SBL for DeepMIMO, compared to ground truth, b) eight exemplary training samples, c) squared absolute value of four exemplary generated samples from all models, respectively.

resulting vectors to channel realizations using the dictionary $\boldsymbol{D}_{t,f}$ and train the autoencoder. We see that by incorporating the prior knowledge that most channels only exhibit a few relevant paths, we improve our method even further and reach almost ground-truth performance.

## F.3. Experiments for the Kronecker Constraint on the Covariances (cf. Appendix C.2)

In Fig. 12, we plot the same results as in Fig. 6, but, additionally, we also plot the performance of the CSGMM and CSVAE with Kronecker-based covariance, explained in Appendix C.2. The Kronecker-based CSVAE does perform similarly or even outperforms the ordinary CSVAE. While the Kronecker-based CSGMM performs well in some configurations, it also exhibits outliers with significant performance drop. As the M-step of the Kronecker-based CSGMM in Appendix C.2 is not the guarenteed global solution of the corresponding optimization problem (39), these performance drops come from the Kronecker-based CSGMM training converging to bad local optima.

### F.4. Experiments for the Parameter Generation on the DeepMIMO Dataset

For the DeepMIMO dataset on SIMO, we evaluate the parameter generation. Equivalent to the results on the modified 3GPP dataset in Fig. 5, we plot the power angular profile and a histogram of the angular spread in Fig. 13 a), exemplary training samples in Fig. 13 b), and the absolute squared value for exemplary newly generated samples of CSVAE, CSGMM, AmbientGAN for $s$ and M-SBL compared to Ground Truth in Fig. 13 c). The results are in line with Fig. 5 with CSGMM exhibiting less peaks and matches the underlying power angular profile more closely. In addition, CSVAE underestimates the amount of LOS channels, which can be seen at the missing peak in the histogram with very small angular spreads. AmbientGAN provides less sparsity than SBGM by not generating parameters with small angular spread.

## G. Pseudocode for Parameter and Channel Generation and Implementation Specifications

Algorithms 1-3 summarize the generation process of parameters, channels, and channels with constraining the path number, respectively. All models and experiments have been implemented in `python 2.1.2` using `pytorch 3.10.13`, and `pytorch-cuda 12.1`. All simulations have been carried out on a `NVIDIA A40 GPU`.

---

**Algorithm 1** Parameter Generation with the CSGMM (*CSVAE*)

---

**Input:** $-$
**Output:** complex-valued vectors $\{s_i\}_{i=1}^{N_{\text{gen}}}$ whose entries together with their indices correspond to the physical parameters (depending on the grid used for training)
**for** $i = 1$ **to** $N_{(\text{gen})}$ **do**
    1) draw $k_i \sim p(k)$ (*or draw $z_i \sim p(z)$*) (cf. Section 4.2)
    2) draw $s_i \sim p(s|k_i) = \mathcal{N}_{\mathbb{C}}(s; \mathbf{0}, \text{diag}(\gamma_i))$ (*or draw $s_i \sim p(s|z_i) = \mathcal{N}_{\mathbb{C}}(s; \mathbf{0}, diag(\gamma_\theta(z_i)))$*) (cf. Section 4.2)
**end for**

---

**Algorithm 2** Channel Generation with the CSGMM (*CSVAE*)

---

**Input:** dictionary $D^{(\text{new})}$ (can be chosen according to the desired system configuration, not necessarily the one used for training)
**Output:** complex-valued channel realizations $\{h_i\}_{i=1}^{N_{\text{gen}}}$ that are scenario-specific and match the desired system configuration of interest
**for** $i = 1$ **to** $N_{(\text{gen})}$ **do**
    1) draw $k_i \sim p(k)$ (*or draw $z_i \sim p(z)$*) (cf. Section 4.2)
    2) draw $s_i \sim p(s|k_i) = \mathcal{N}_{\mathbb{C}}(s; \mathbf{0}, \text{diag}(\gamma_i))$ (*or draw $s_i \sim p(s|z_i) = \mathcal{N}_{\mathbb{C}}(s; \mathbf{0}, diag(\gamma_\theta(z_i)))$*) (cf. Section 4.2)
    3) compute $h_i = D^{(\text{new})} s_i$
**end for**

---

**Algorithm 3** Channel Generation with the CSGMM (*CSVAE*) when only considering $p_{\max}$ paths

---

**Input:** dictionary $D^{(\text{new})}$ (can be chosen according to the desired system configuration, not necessarily the one used for training)
**Output:** complex-valued channel realizations $\{h_i\}_{i=1}^{N_{\text{gen}}}$ that are scenario-specific and match the desired system configuration of interest
**for** $i = 1$ **to** $N_{(\text{gen})}$ **do**
    1) draw $k_i \sim p(k)$ (*or draw $z_i \sim p(z)$*) (cf. Section 4.2)
    2) draw $s_i \sim p(s|k_i) = \mathcal{N}_{\mathbb{C}}(s; \mathbf{0}, \text{diag}(\gamma_i))$ (*or draw $s_i \sim p(s|z_i) = \mathcal{N}_{\mathbb{C}}(s; \mathbf{0}, diag(\gamma_\theta(z_i)))$*) (cf. Section 4.2)
    3) compute $|s_i|^2$ and identify the indices $\mathcal{I}_i$ of the $p_{\max}$ strongest entries in $|s_i|^2$
    4) compute $\tilde{s}_i$ which equals $s_i$ for the indices in $\mathcal{I}_i$ and is zero elsewhere
    3) compute $h_i = D^{(\text{new})} \tilde{s}_i$
**end for**

---

