# OpenReview forum: "Physics-Informed Generative Modeling of Wireless Channels"
_ICML.cc/2025/Conference — ICML 2025 poster_

### Official Review · Reviewer_BGie · 2025-03-11

**Overall Recommendation:** 3

**Summary:**

This paper introduces a physics-informed generative model for wireless channel modeling, addressing challenges in data efficiency, generalizability, and physical interpretability. Unlike traditional stochastic models or recent GAN-based methods, the authors propose a Sparse Bayesian Generative Modeling (SBGM) framework that leverages the compressibility of wireless channels to learn distributions of physical channel parameters (e.g., delays, angles of arrival/departure). The results demonstrate that the proposed approach produces physically interpretable channel models that maintain statistical and structural consistency with real-world wireless propagation environments.

**Claims And Evidence:**

Good to me.

**Essential References Not Discussed:**

None

**Experimental Designs Or Analyses:**

The experimental setup is comprehensive, covering a range of scenarios and datasets to validate the proposed model. The authors conduct experiments on standardized 3GPP spatial channel models, QuaDRiGa simulations, and DeepMIMO ray-tracing datasets, ensuring that their results are applicable to both empirical and simulated wireless environments. The evaluation spans different system configurations, including single-input-multiple-output (SIMO) and orthogonal frequency-division multiplexing (OFDM), demonstrating the flexibility of the approach.

A notable aspect of the experimental design is the inclusion of cross-validation using an autoencoder-based reconstruction method, which provides a robust metric for assessing the realism of generated channels. The authors also perform sensitivity analyses, varying the number of training samples and pilot symbols to examine the model’s robustness. One limitation of the experimental design is the lack of ablation studies to isolate the contributions of different components, such as the impact of sparsity constraints or the Kronecker factorization. Additionally, a discussion on training stability and sensitivity to hyperparameters would further improve the evaluation.

**Methods And Evaluation Criteria:**

The authors evaluate their methods on three different dataset: a modified standardized 3GPP spatial channel model for SIMO for a better illustration; QuaDRiGa for simulations with OFDM; and a further evaluation on the ray tracing database DeepMIMO in Appendix L. It justify the physical parameter distribution accuracy by the comparison of generated vs. real distributions for angular spread, power angular profile, and delay-Doppler characteristics. The channel generation performance is evaluated using cross-validation autoencoder reconstruction error (normalized MSE and cosine similarity). It demonstrates it outperformance by the comparisons with GAN-based baselines (AmbientGAN, GAN-rbf). Overall, these metrics align well with the goals of wireless generative modeling and provide strong empirical validation.

**Other Comments Or Suggestions:**

Good to me.

**Other Strengths And Weaknesses:**

One of the major strengths of this paper is its ability to generate physically consistent channel models while maintaining computational efficiency. The approach is novel in that it shifts from implicit GAN-based modeling to a prescribed statistical framework that ensures generalizability across system configurations. The integration of structured priors further enhances sample efficiency and interpretability.

However, the paper lacks ablation studies that isolate the effects of different components, such as the impact of sparsity constraints, Kronecker factorization, and dictionary-based representations. Additionally, while the method generalizes across system configurations, there is limited discussion on generalization to different propagation environments or frequency bands.

**Questions For Authors:**

1. How well does the model generalize to unseen frequency bands or new environments?
2. How sensitive is SBGM to different levels of noise or missing data?

**Relation To Broader Scientific Literature:**

The paper bridges physics-based wireless models and ML-driven generative methods, ensuring interpretability and generalizability. Unlike GANs, it leverages sparse Bayesian learning and structured priors, aligning with trends in physics-informed ML.

**Theoretical Claims:**

Good to me.

---

> ### Author Rebuttal · Authors · 2025-03-31
>
> We thank reviewer BGie for the reviewer's comments. In the following, we address the reviewer's questions and concerns.
>
> **To Other Strengths And Weaknesses (2. paragraph):** We agree with the reviewer that further ablations studies regarding various hyperparameters would improve our work's quality. Therefore, we conducted preliminary experiments regarding the number K of CP-GMM components as well as our models' grid resolution. More specifically, in Fig. 2 in the anonymous repository https://github.com/anonymousicml2529/rebuttal, we plot the angular spread for different numbers K of CP-GMM components for the modified 3GPP dataset. For K = 1, the CP-GMM coincides with M-SBL [1]. It can be seen that for K<4, the angular spread is significantly overestimated, which is to be expected as the underlying data is coming from a scenario with power from four different angular regions (cf. Fig. 3 in Appendix H). In Fig. 3 (https://github.com/anonymousicml2529/rebuttal), we investigate different grid resolutions for the QuaDRiGa 5G OFDM dataset. In these simulations, we vary the resolution of the delay-doppler grid with $S^2$ gridpoints from $S=16$ to $S=64$, while keeping the maximally resolvable delay at $8\mu\text{s}$ and the maximally resolvable doppler at $100\text{Hz}$. It can be seen that the grid resolution has only marginal effects on the performance of the autoencoder in a) and b). In c), we plot exemplary samples from the trained CP-GMMs, respectively.
>
> **To Questions for Authors (1):** There are various effects influencing the channel when considering a different frequency band. Interestingly, for spatial channels, we can adapt to different frequency bands by considering a new dictionary $\mathbf{D}$, that takes the newly considered frequency band into account. More precisely, the so-called beam squint (cf. [2]) is the major cause for changed spatial channel characteristics when varying frequency bands. The beam squint describes that the spatial steering vectors are frequency-dependent as they depend on the ratio between the antenna spacing and the considered wavelength. However, as the steering vectors constitute the dictionary $\mathbf{D}$ in our model, we can adapt this ratio in a newly used dictionary after training. The remaining frequency-dependent effects regarding, e.g., the path loss are rather slowly changing, which is why we expect our model to be valid for a reasonably broad frequency range. We will consider including experiments regarding this in our work.
>
> We would like to mention that our model is supposed to capture the scenario-specific channel characteristics from one concrete environment. Only then, it can generate new samples that are better suited for this environment in comparison to generic standardized channel models (e.g., 3GPP). A generalization for a large variety of environments is what the already existing standardized channel models are designed for. This is why we do **not** want our model to generalize very well to unseen environments, as this would likely lead to a considerable drop in performance for the environment it has been trained for. Of course, there is a certain trade-off that can be optimized. However, we do not think that this is within the scope of our work, and we hope that the reviewer agrees with us on this point.
>
> **To Questions for Authors (2):** In our simulations, we added noise to our training samples whose noise variance is drawn such that the corresponding SNR is uniformly distributed between 0 and 20 dB. Moreover, SBGM is specifically designed to deal with noisy and corrupted data and considers the noise level during training. Therefore, SBGM is robust with respect to different noise levels. In our OFDM simulations, we consider solely training samples that contain missing data, and our model performs very well. As a consequence, SBGM also works well for missing data. We hope that this is what the reviewer is referring to and we did not misunderstand the reviewer's question. If not, we would appreciate it if the reviewer could comment again on this point.
>
> We hope that we addressed all the reviewer's concerns and would like to thank the reviewer again, as we think that the additional experiments regarding the hyperparameters will improve our work's quality.
>
> [1] Wipf et al., ”An Empirical Bayesian Strategy for Solving the Simultaneous Sparse Approximation Problem,” IEEE TSP., vol. 55
>
> [2] B. Wang, M. Jian, F. Gao, G. Y. Li and H. Lin, "Beam Squint and Channel Estimation for Wideband mmWave Massive MIMO-OFDM Systems," in IEEE Transactions on Signal Processing, vol. 67

---

### Official Review · Reviewer_dNNa · 2025-03-14

**Overall Recommendation:** 4

**Summary:**

The paper presents a new way to model wireless channels by combining basic physical principles with modern generative modeling techniques. Instead of using complex black-box methods like GANs—which require lots of clean data—the authors introduce more structured approaches based on simplified versions of VAEs and GMMs (called CP‑VAE and CP‑GMM). These models learn the key physical characteristics of wireless channels (such as the angles, delays, and Doppler shifts) even when the data is noisy or incomplete. By incorporating known physics rules into the learning process, the method produces realistic and interpretable channel models. Additionally, the approach uses clever algorithmic tricks (like a closed-form update using Kronecker factorization) to reduce computational effort and can adapt to various wireless setups without needing to start over with training. Tests on several datasets show that this new method generates accurate channel representations and outperforms traditional GAN-based models.

## Update After Rebuttal:
The authors sufficiently address my review and I vote to keep the score.

**Claims And Evidence:**

Most of the paper’s claims are supported by a mix of theoretical derivations and experimental results on simulated datasets.

**Essential References Not Discussed:**

The related work in this paper appears to be appropriate for the paper.

**Experimental Designs Or Analyses:**

The paper runs experiments on simulated data. While simulated datasets are standard, the experimental validation does not include real-world measurement data. Testing on real-world channels would further confirm the practical viability of the approach.

 Details on hyperparameter selection and sensitivity analysis are limited. More thorough reporting on these aspects could help validate the robustness of the methods.

**Methods And Evaluation Criteria:**

## Methods

- **CP-VAE and CP-GMM Models:**
  - Special versions of common generative models (variational autoencoders and Gaussian mixture models).
  - Leverage physical properties of wireless channels (like compressibility and regular covariance patterns) to learn key parameters (angles, delays, Doppler shifts) even from noisy or incomplete data.
  - Designed for common wireless setups such as SIMO and OFDM.
  - Can adapt to different system configurations without needing to retrain.

## Evaluation and Datasets

- **Datasets Used:**
  - A modified 3GPP model for SIMO channels.
  - QuaDRiGa for OFDM channels.
  - The DeepMIMO ray tracing database.

- **Evaluation Metrics:**
  - **Power Angular Profile**
  - **Angular Spread**
  - **Normalized Mean Squared Error (NMSE)**
  - **Cosine Similarity**

Overall, these methods and evaluation criteria are well-suited for capturing and validating the real-world behavior of wireless channels.

**Other Comments Or Suggestions:**

None

**Other Strengths And Weaknesses:**

## Strengths
- **Originality:**
  - Integrates physical channel properties with sparse Bayesian generative modeling.
  - Introduces novel prescribed models (CP-VAE and CP-GMM) that reduce the need for high-quality data.
- **Significance:**
  - Addresses key challenges in wireless channel modeling, enhancing interpretability and adaptability.
  - Demonstrates potential for use across various system configurations without retraining.
- **Clarity:**
  - Overall presentation is clear with well-defined equations and detailed explanations.

## Weaknesses
- **Experimental Validation:**
  - Evaluation is primarily based on simulated datasets; real-world datasets might further strengthen the claims.

**Questions For Authors:**

1. **Real-World Data Validation:**
   Could you clarify whether you have tested your method on real-world channel measurements, and if not, what are your expectations or plans for such validation?
   *If the authors provide evidence or detailed plans that indicate strong performance on real-world data, it would significantly reinforce the practical impact of the work.*

2. **Hyperparameter Sensitivity and Robustness:**
   How sensitive is your approach to the choice of hyperparameters (e.g., grid resolution, noise variance estimates) and what guidelines can you offer for setting them in different scenarios?
   *A detailed sensitivity analysis or robust guidelines could increase confidence in the method’s reproducibility and general applicability.*

**Relation To Broader Scientific Literature:**

The paper builds on ideas from sparse Bayesian learning and compressive sensing (as developed by Tipping (2001) and Wipf & Rao (2004)), adapting them to the wireless channel context. It leverages SBGM, previously explored in works like Böck et al. (2024b), to derive models (CP‑VAE and CP‑GMM) that are both data-efficient and physically interpretable. The use of standard datasets like QuaDRiGa and DeepMIMO situates the work within the broader literature, allowing direct comparison with existing channel models.

**Theoretical Claims:**

There are no theoretical proofs in the first 9 pages, and I did not read them.

---

> ### Author Rebuttal · Authors · 2025-03-31
>
> We want to thank reviewer dNNa for the thorough review and the reviewer's assessment. In the following, we address the reviewer's additional questions.
>
> **To Experimental Designs Or Analyses (1. paragraph) & Weaknesses & Questions For Authors (1):** We agree with the reviewer that experimental results on real measurement data would strengthen our paper's contribution. We indeed have preliminary results of our models on a real-world measurement campaign that was conducted on the Nokia campus in Stuttgart, Germany, in 2017. In Fig. 4 in the anonymous repository https://github.com/anonymousicml2529/rebuttal, you can see the scenario in which measurements have been conducted. Moreover, the green regions indicate that users experienced a LoS path to the base station, while the red regions have solely NLoS paths. The base station was equipped with a 16x4 antenna array. In our preliminary results, we only consider the first row of the antenna array. In Fig. 5 (https://github.com/anonymousicml2529/rebuttal), we tested the CP-VAE on this data. The data consisted of 16-dimensional noisy channel measurements in the spatial domain. In a), you can see the overall power angular profile of newly generated samples (256-dimensional). In Fig. b), we trained the CP-VAE solely on LoS channels from the "LoS"-street canyon, as well as solely on NLoS channels respectively. It can be seen that the generated channel parameters from the LoS CP-VAE exhibit smaller angular spreads, which is expected. Moreover, in c), you can see exemplary plotted samples validating that the CP-VAE can easily distinguish between the LoS and NLoS channels. We will consider including these results in our work.
>
> **To Experimental Designs Or Analyses (2. paragraph) & Questions For Authors (2):** We agree with the reviewer that a further analysis regarding the hyperparameters would improve our work's contribution. In Fig. 2 and 3 in https://github.com/anonymousicml2529/rebuttal, we have preliminary results for these experiments.
>
> More precisely, in Fig. 2, we investigate the effect of the CP-GMM's number of components on the angular spread for the modified 3GPP data. For K=1, the CP-GMM coincides with M-SBL [1]. It can be seen that for K<4, the angular spread is significantly overestimated, which is to be expected as the underlying data is coming from a scenario with power from four different angular regions (cf. Fig. 3 in Appendix H).
>
> In Fig. 3, we investigate different grid resolutions for the QuaDRiGa 5G OFDM dataset. In these simulations, we vary the resolution of the delay-doppler grid with $S^2$ gridpoints from $S=16$ to $S=64$, while keeping the maximally resolvable delay at $8\mu\text{s}$ and the maximally resolvable doppler at $100\text{Hz}$. It can be seen that the grid resolution has only marginal effects on the performance of the autoencoder in a) and b). In c), we plot exemplary samples from the trained CP-GMMs, respectively.
>
> In our final version, we will consider including a more thorough discussion and experimental validation regarding various hyperparameters, such as the number K of CP-GMM components, the grid-resolution, and the maximally resolvable delay and doppler. Moreover, we will consider including some results for the measurement campaign. We would like to thank the reviewer dNNa again as we think that both remarks will improve our work's quality.
>
> [1] Wipf et al., ”An Empirical Bayesian Strategy for Solving the Simultaneous Sparse Approximation Problem,” IEEE TSP., vol. 55

---

### Official Review · Reviewer_9VKv · 2025-03-16

**Overall Recommendation:** 4

**Summary:**

This paper introduces a physics-informed generative modeling approach for wireless channels. By using sparse Bayesian generative modeling and knowledge about the conditional channel moments, this paper addresses limitations in existing methods such as the need for high-quality data, lack of generalizability, and poor physical interpretability.

**Claims And Evidence:**

The claims are well supported.

**Essential References Not Discussed:**

No.

**Experimental Designs Or Analyses:**

This paper made experiments for both SIMO and OFDM, and used open datasets to evaluate the proposed method. The experimental designs are sound and the evaluation metrics are well chosen.

**Methods And Evaluation Criteria:**

This paper figures out a way to solve the drawback of sparse Bayesian generative modeling by using the conditional channel moments.

**Other Comments Or Suggestions:**

Not appliable.

**Other Strengths And Weaknesses:**

Strengths: The paper provides a data-knowledge dual-driven way to model the wireless channel, which is important for the simulations in communication and signal processing. According to the experiments, the proposed methods present a superior performance over the other methods.

Weakness:
- The paper seems more suitable for communication/signal processing in my opinion.
- It would be better if the paper could compare with more other methods.

**Questions For Authors:**

1. How do you choose the selection matrix "A"? Do you choose a good one by cross-correction?
2. What if the matrix “A” changes for different training channels?

**Relation To Broader Scientific Literature:**

The contributions of this paper is useful for designing algorithms for communication and signal processing, but I don't realize the contribution to machine learning.

**Theoretical Claims:**

Not appliable.

---

> ### Author Rebuttal · Authors · 2025-03-30
>
> We thank reviewer 9VKv for the review and the reviewer’s assessment. In the following, we address the concerns and questions raised.
>
> **To Relation To Broader Scientific Literature & Other Strengths and Weaknesses (bullet point 1):** We agree that our proposed model can be utilized in the context of communications and signal processing. Having said that, we also strongly believe that our work fits well with the application-driven machine-learning track of ICML. Our model builds a machine learning-based framework to simulate the propagation of electromagnetic waves in indoor and outdoor scenarios. For that, we enhance a machine learning-based framework (sparse Bayesian generative modeling) by incorporating specific insights from geometric optics and electromagnetic waves (i.e., the channel sparsity and the structural knowledge about conditional channel moments). In this sense, we derive a new application-driven machine learning-based tool that is not meant to solve one particular problem from communications or signal processing but can be used generally. We hope that the reviewer agrees with us on this.
>
> **To Other Strengths And Weaknesses (bullet point 2):** We now included two more baselines for physical parameter generation: an AmbientGAN that does not output channel realizations $\mathbf{h}$ but their compressible representation $\mathbf{s}$, and M-SBL from [1]. In Fig. 1 in the anonymous repository https://github.com/anonymousicml2529/rebuttal/, we plot preliminary results for the 3GPP dataset. It can be seen that AmbientGAN, as well as M-SBL, significantly overestimate the channel-wise angular spread. We would also like to refer to our response *To Experimental Designs or Analyses (1. paragraph)* to Reviewer 1 (9aFT), where we discuss the new baselines and the results in more detail.
>
> **To Questions For Authors (1)):** In our simulations for OFDM, $\mathbf{A}$ is chosen as a selection/masking matrix, i.e., it masks out a certain amount of entries in the corresponding channel. This is in line with the OFDM standard in wireless communications. The indices that correspond to the entries that are masked out are drawn randomly. Moreover, this masking operation is illustrated in, e.g., Fig. 2 e), where we plot the corresponding compressed training samples.
>
> **To Questions For Authors (2)):** Changing the observation matrix for different training samples is possible and has also been considered in [2]. Generally, changing the observation matrix either has no impact or even improves the overall performance as the model experiences varying ”subspace information” from the training samples during training. In our work, we only considered fixed measurement matrices as this is arguably more relevant for wireless channels. However, we will consider including simulations regarding varying the observation matrix in the final version.
>
> We would like to thank the reviewer 9VKv again for the valuable comments, and we think that the added baselines will improve our work's quality.
>
> [1] Wipf et al., ”An Empirical Bayesian Strategy for Solving the Simultaneous Sparse Approximation Problem,” IEEE TSP., vol. 55
>
> [2] Böck et al., ”Sparse Bayesian generative modeling for compressive sensing,” NeurIPS, 24

---

### Official Review · Reviewer_9aFT · 2025-03-19

**Overall Recommendation:** 3

**Summary:**

The paper proposes a physics‐informed generative modeling framework for wireless channels that integrates physical channel knowledge with sparse Bayesian generative modeling (SBGM). The method consists of several key components:

Channel Representation via Compressibility:
The authors begin by exploiting the fact that wireless channels are compressible when expressed in a physically interpretable dictionary.
In an SIMO channel, the representation is a sparse vector that captures the physical parameters (such as delays and angles). A similar representation is used for OFDM channels by defining a grid in the delay–Doppler domain.

Sparse Bayesian Generative Modeling (SBGM):
Building on SBGM, the method learns a parameterized distribution for the compressible representation of the channel.

Incorporating Physical Structure:
The model leverages known physical properties of wireless channels. Because path loss phases are uniformly distributed, channels tend to be zero-mean and, under stationarity assumptions, have structured (e.g., Toeplitz) covariance matrices. By choosing the dictionary  appropriately, the conditional covariance naturally inherits a Toeplitz (or block-Toeplitz) structure. This aligns the learned model with established physical channel models.

Model Variants – CP-VAE and CP-GMM:
Two concrete implementations are proposed:
Channel Parameter-VAE (CP-VAE): A variational autoencoder variant that learns the latent distribution and uses neural networks to parameterize $\gamma_\theta (z)$.
Channel Parameter-GMM (CP-GMM): A Gaussian mixture model version where the latent variable is discrete and the model learns mixture weights and associated diagonal covariance matrices.
Both variants are designed to capture the distribution of the compressible representations, which in turn determine the physical channel parameters.

Kronecker Approximation for High-Dimensional Channels:
In scenarios such as OFDM where the channel spans multiple domains (e.g., time and frequency), the model incorporates a Kronecker structure to the conditional covariance. This reduces the number of parameters by expressing the covariance as a Kronecker product of two smaller Toeplitz matrices, making the method more scalable.

Generalizability and Data Efficiency:
A notable aspect of the approach is its ability to learn from compressed and noisy observations (e.g., those collected during standard online operation) rather than requiring extensive high-quality training data. Furthermore, the learned generative model is shown to generalize to different system configurations (such as varying numbers of antennas or subcarrier spacings) by adapting the dictionary without retraining.

**Claims And Evidence:**

The evidence for the claims are reasonable. The authors validate their generative model by studying the power angular profile and channel-wise angle spread. They also use the channels generated by their methods and baselines to train autoencoders. The reconstruction error of these autoencoders are then evaluated, and the proposed method performs better.

**Essential References Not Discussed:**

Seems reasonable.

**Experimental Designs Or Analyses:**

The benchmark algorithms seem weak. AmbientGAN is work from 2018 and WGAN-rbf seems like a method the authors made up (it does not contain a citation). Surely there must be more recent algorithms for learning wireless channels (I'm not an expert on wireless channel generation, so I cannot provide feedback on appropriate algorithms). Why not compare to the algorithms by Baur et al and Fesl et al from the related works section?

Another reasonable evaluation would be on the performance of these models for compressed sensing estimation-- the current work is evaluate on reconstruction of fully observed channels, and I did not find experiments for estimation from compressed measurements.

**Methods And Evaluation Criteria:**

The method and evaluation datasets make sense, although there is confusion about what the contributions of this work are in comparison to prior work.

The paper is written in an extremely confusing manner. The authors start section 4.1 with sparse modelling of wireless networks (which seems fairly standard and I don't understand what the innovation is here), then move to sparse generative modelling (again, seems like prior work from Wipf and Rao directly transfer), then move to section 4.3 where they talk about how wireless channels don't suffer the same problems suffered by other sparse dictionaries, and the techniques from Bock et al directly transfer with some modifications. There's too much unnecessary text that blurs the actual method proposed by the authors and what their contributions are. The introduction and background material take up almost 4.5 pages, which is far too much IMO.

There's also no clear reasoning as to why this algorithm needs fewer training samples than existing work. There are no experiments that study the effect of training size on the proposed algorithm and baselines.

**Other Comments Or Suggestions:**

N/A

**Other Strengths And Weaknesses:**

N/A

**Questions For Authors:**

My main concerns are regarding the writing/description of the algorithm. A more detailed description of why your method is not a direction extension over the prior works by Bock et al, is appreciated.

There's also no clear reasoning as to why this algorithm needs fewer training samples than existing work. There are no experiments that study the effect of training size on the proposed algorithm and baselines.

Additionally, the performance of these models on compressed sensing estimation seems like the most relevant metric for judging the quality of these models.

**Relation To Broader Scientific Literature:**

The paper seems to be a significant contribution towards the generative modelling of parameter-based wireless channels. However, there are some concerns regarding the novelty over prior work by Bock et al. My reading of the paper is that most of the techniques for sparse generative modelling transfer to this case, with some modifications for wireless channels.

**Theoretical Claims:**

The paper does not report theoretical results.

---

> ### Author Rebuttal · Authors · 2025-03-30
>
> We thank reviewer 9aFT for the thorough summary and review. In the following, we address the concerns and questions raised.
>
> **To Methods and Evaluation Criteria (2. paragraph) & Relation to Broader Scientific Literature & Questions for Authors (1. paragraph):** We agree with the reviewer that Section 4 contains a mix of our contributions and already existing ideas and techniques, such as channel sparsity, sparse Bayesian generative modeling, and knowledge of conditioned channel moments. We decided for this structure since our main contribution lies in recognizing that these topics can be combined to yield a generative model for wireless channels that is very different from existing approaches and overcomes the main concerns in concurrent methods. However, we understand that this might make extracting our contribution difficult.
>
> From our perspective, our work has considerable application-driven contributions: It addresses and overcomes the major concerns of state-of-the-art generative models for wireless channels by combining three topics that are non-trivial to combine, i.e., channel sparsity, sparse Bayesian generative modeling (which is originally not meant to generate samples), and prior knowledge of conditioned channel moments. By doing so, we derive a model that can train on corrupted data, is physically interpretable, and requires no retraining for changing system configurations. In addition, our work has a methodological contribution by deriving a new EM algorithm in Appendix F that exploits the conditioned Toeplitz structure.
>
> We will rewrite the final version so that our contribution is better separated from previous work. We thank the reviewer for making us aware of this potentially misleading structure.
>
> **To Methods and Evaluation Criteria (3. paragraph) & Questions for Authors (2. paragraph):** We would like to refer to Fig. 2 a) and b), where we investigate the effect of the training size. In the considered QuaDRiGa scenario, 100 training samples are sufficient for our method to perform well.
>
> **To Experimental Designs or Analyses (1. paragraph):** Generative modeling for wireless channels is rather new, and there are basically no proposed methods that learn from corrupted data. Almost all approaches are based on GAN-variants and assume lots of ground-truth training data. Although the methods in Baur et al. and Fesl et al. can learn from compressed data, they are not derived to generate samples but directly apply to estimating wireless channels from observations.
>
> Having said that, we included two more baselines now. In our submission, we compared to AmbientGAN, which can output channels h. However, AmbientGAN can also be trained to output the compressible channel representation s by including the dictionary D between generator and discriminator. This also allows us to compare to AmbientGAN for parameter generation. Moreover, we included M-SBL [1]. M-SBL is an SBL extension that assumes several observations instead of one. Indeed, M-SBL is equivalent to the CSGMM from [2] with only one component. In Fig. 1 in the anonymous repository https://github.com/anonymousicml2529/rebuttal/, we plot preliminary results for the 3GPP dataset. While AmbientGAN for s and M-SBL perform similarly for the power angular profile, both significantly overestimate the channel-wise angular spread. For AmbientGAN, this is due to the lack of sparsity prior knowledge. For M-SBL, it is because it fits a Gaussian to s, which is not expressible enough to capture different directions individually. This is also seen in the plotted samples (c).
>
> **To Experimental Designs or Analyses (2. paragraph) & Questions For Authors (3. paragraph):** There might be a misunderstanding regarding our evaluation method. We do **not** evaluate the reconstruction performance of our method in any experiments. An autoencoder reconstructs generated data as a means to evaluate the **generation** performance of our model. Specifically, the autoencoder is first trained on data generated by our model. It then reconstructs ground-truth channels. As our model generates new channels that are supposed to resemble ground truth, we do not think that training the autoencoder on compressed data is a good fit for evaluation. Applying our model to compressed sensing, as in [2], evaluates the performance for reconstruction. However, we consider the generation performance, so we evaluate the quality of newly generated samples. If the reviewer is referring to something different and we misinterpret the statement, we would appreciate it if the reviewer could comment on this point again.
>
> We would like to thank the reviewer again as we think that restructuring our work together with the newly added baselines will improve our work's quality.
>
> [1] Wipf et al., ”An Empirical Bayesian Strategy for Solving the Simultaneous Sparse Approximation Problem,” IEEE TSP., vol. 55
>
> [2] Böck et al., ”Sparse Bayesian generative modeling for compressive sensing,” NeurIPS, 24

---

### Decision · Program_Chairs · 2025-05-01

**Decision:**

Accept (poster)

**Comment:**

This submission presents a novel integration of physical channel properties with sparse Bayesian generative modeling (SBGM) to address key challenges in wireless channel modeling. The proposed CP-VAE and CP-GMM frameworks aim to improve data efficiency, generalizability, and physical interpretability compared to GAN-based approaches.

Based on the reviews, I would like to recommend acceptance due to the following strengths of this submission:

1. The paper addresses a timely challenge in wireless communications with a technically sound, physics-informed approach.

2. Author revisions substantially improved clarity, baseline comparisons, and empirical validation (including real-world data).


Nevertheless, as also pointed out by reviewers, some improvement can be made as follows:

1. Explicitly highlight methodological novelties (e.g., EM algorithm for Toeplitz structure) in the introduction.

2. Include real-world measurement results (Nokia dataset) in the main text.

3. Discuss limitations regarding frequency band generalization.